# SCENARIO-WISE REC: A MULTI-SCENARIO RECOMMENDATION BENCHMARK

## ABSTRACT

Multi Scenario Recommendation (MSR) tasks, referring to building a unified model to enhance performance across all recommendation scenarios, have recently gained much attention. However, current research in MSR faces two significant challenges that hinder the field's development: the absence of uniform procedures for multi-scenario dataset processing, thus hindering fair comparisons, and most models being closed-sourced, which complicates comparisons with current SOTA models. Consequently, we introduce our benchmark, **Scenario-Wise Rec**, which comprises 6 public datasets and 12 benchmark models, along with a training and evaluation pipeline. Additionally, we validated the benchmark using an industrial advertising dataset, reinforcing its reliability and applicability in real-world scenarios. We aim for this benchmark to offer researchers valuable insights from prior work, enabling the development of novel models based on our benchmark and thereby fostering a collaborative research ecosystem in MSR. Our source code is also publicly available[1].

## 1 INTRODUCTION

Recommender systems, deeply integrated into the digital world, play a crucial role in mitigating data overload and personalizing user experiences across diverse online platforms (Zhang et al., 2019; Fan et al., 2022; Zhang et al., 2021). Current recommender systems leverage user profiles, behavior sequences, and contextual features to produce customized recommendations for specific user and item scenarios (Zhou et al., 2019). In the face of varied real-world applications, there is growing research on the development of models capable of managing multiple recommendation scenarios simultaneously, known as the Multi-Scenario Recommendation (MSR) task. MSR models, tailored to unique user and item scenarios, dynamically learn to transfer knowledge across scenarios (also referred to as "domains" in some research). This strategy not only addresses data scarcity in less populated scenarios but enhances overall recommendation performance (Feng et al., 2020; Xie et al., 2022).

Specifically, multi-scenario recommendations involve designing a unified model capable of generating recommendations across multiple scenarios (Sheng et al., 2021; Yang et al., 2022; Wang et al., 2022). These scenarios often represent distinct predefined domains, such as various advertising areas, product pages, or manually defined business units shown in Figure 1. The model's primary objective is to harness knowledge transfer across scenarios to improve scenario-specific performance. Central to these models is the ability to balance shared information and specific information across different scenarios, thereby enhancing the overall predictive accuracy. This capability is especially crucial for real-life deployments, where enterprises frequently face the challenge of executing recommendation tasks across multiple scenarios (Zhang et al., 2022).

With the development of deep recommender systems (Zhang et al., 2019; Batmaz et al., 2019) and cross-domain studies (Zhu et al., 2021a; Gao et al., 2023), we have witnessed the rapid growth of multi-scenario recommendation methods. Many models, such as STAR (Sheng et al., 2021), AdaSparse (Yang et al., 2022), PEPNet (Chang et al., 2023), ADL (Li et al., 2023a), $M^3$oE (Zhang et al., 2024), among others, have been proposed and effectively implemented. However, there is still a lack of a widely universally recognized benchmark in this area, which poses significant challenges:

---

[1]https://anonymous.4open.science/r/Scenario-Wise-Rec-05B5

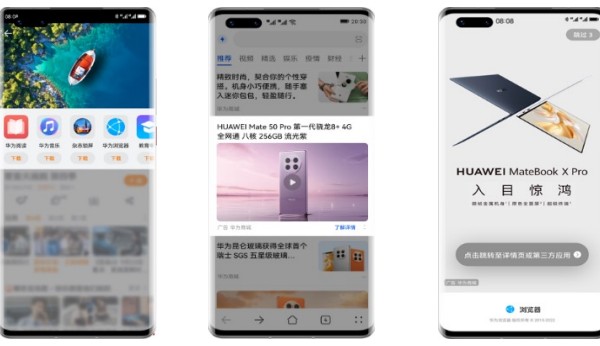

(a) App Icon Slot     (b) Stream Video Slot     (c) Open Screen Slot

Figure 1: An MSR example in business application: multi-scenario advertising recommendations. Each slot is treated as a specific scenario in modeling.

Firstly, there is a lack of a standardized pipeline for scenario data processing, model training, and model performance evaluation to make fair comparisons between models. Secondly, many current MSR models are closed-sourced due to corporate privacy protection policies, which complicates reproducibility for researchers, thereby impeding the field's progression in multi-scenario recommendations.

Given these challenges, the demand for a well-defined benchmark, specifically tailored for multi-scenario recommendations, grows increasingly urgent. This benchmark should provide standardized procedures for data processing, evaluation, and model interfaces, thereby establishing uniform research norms. In this paper, we propose **Scenario-Wise Rec**, the first benchmark dedicated to MSR. Our benchmark incorporates data preprocessing and evaluation protocols for six public scenario datasets, providing a structured framework for model comparison and ensuring equitable evaluation conditions. We have developed a uniform model interface and reproduced ten well-recognized MSR models, including three multi-task-related models and seven multi-scenario models. To validate our benchmark's applicability and robustness, we have also applied it to an industrial dataset from one online advertising platform, demonstrating its real-world performance. Our comprehensive approach not only enables researchers to derive valuable insights from existing works but also aims to nurture a collaborative research environment within the MSR field. The main contribution could be listed as follows:

- To the best of our knowledge, this is the first open-source benchmark designed for cutting-edge MSR research, incorporating the latest models and a diverse MSR datasets. It serves the needs of both academic and industrial research communities, bridging the gap between the latest advancements in both fields.

- Our benchmark offers a unified pipeline for MSR tasks, covering data preprocessing, model training, and evaluation. integrating six public datasets and twelve widely recognized MSR models for fair comparisons and reproducibility. Additionally, the benchmark is validated with an industrial advertising dataset, enhancing its credibility and real-world applicability.

- We have made our benchmark publicly accessible, enabling researchers to conduct MSR experiments with ease and gain valuable insights. This initiative aims to simplify MSR experimentation, foster collaboration, and accelerate progress within the MSR community.

## 2 RELATED WORK

In recent years, interest in multi-scenario recommendation tasks has surged, driven by the rapid growth in user numbers and web content. Platform providers segment user groups and content themes into distinct scenarios based on different kinds of recommendation needs (E.g., different advertising slots), resembling multi-task learning. Researchers have been exploring scenario-transfer technologies to address these challenges. Notable efforts are introduced which use Mixture-of-Expert (MoE) structures to manage scenario diversity. Mario (Tian et al., 2023) captures scenario

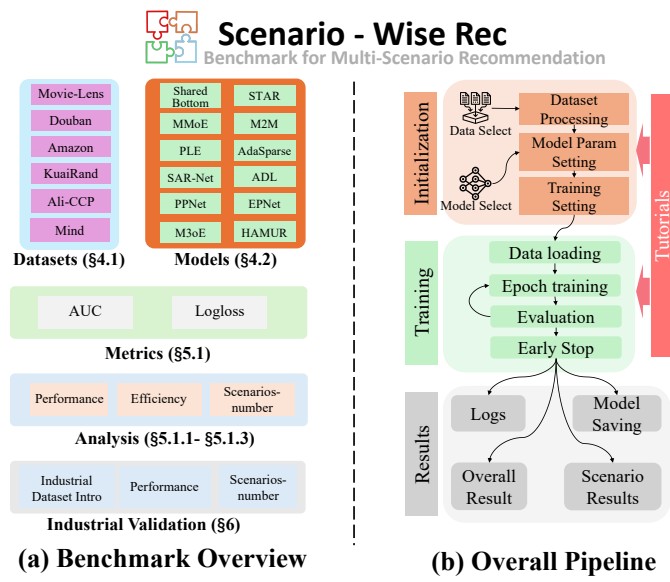

Figure 2: Overall pipeline of Scenario-Wise Rec.

information through feature scaling modules and dynamically uses MoE structures. HiNet (Zhou et al., 2023) uses hierarchical structures for effective scenario information extraction while preserving scenario-specific features. PEPnet (Chang et al., 2023) employs gating units for bottom-level input processing and introduces EPNet for scenario feature selection and PPNet for integrating multi-task information. Other approaches address scenario modeling differently. STAR (Sheng et al., 2021) introduces a unified model with scenario-specific and scenario-shared towers to capture unique and shared information. SAR-Net (Shen et al., 2021) and SAML (Chen et al., 2020) use attention mechanisms for scenario feature modeling, facilitating knowledge transfer and improving performance. ADL (Li et al., 2023a) distinguishes scenario communities through an adaptation module, and other research explores scenario knowledge transfer via embedding alignment. CausalInt (Wang et al., 2022) uses causal inference for multi-scenario recommendations, and AdaSparse (Yang et al., 2022) applies pruning strategies for scenario adaptation.

Recent studies include HAMUR (Li et al., 2023b), which utilizes scenario adapters for improved distribution adaptation, and PLATE (Wang et al., 2023), which employs prompt technology for scenario adaptation. D3 (Jia et al., 2024) focuses on autonomous scenario-splitting, while MDRAU (Ju et al., 2024) leverages "seen" scenarios to address "unseen" ones. M-scan (Zhu et al., 2024) introduces a Scenario-Aware Co-Attention mechanism and a Scenario Bias Eliminator. Additionally, Uni-CTR (Fu et al., 2023) uses LLMs to extract semantic representations across scenarios in MSR, and M³oE (Zhang et al., 2024) refines Mixture-of-Expert (MoE) modules, extending them for multi-scenario and multi-task settings. Our benchmark systematically summarizes the MSR task, offering a comprehensive pipeline that includes datasets, models, training processes, and evaluation, providing researchers with a solid foundation for further exploration in this field.

## 3 PIPELINE

In this section, a detailed introduction to the components of our benchmark is given, the overview framework is shown in Figure 2.

- **Task: Multi-scenario Click-Through Rate Prediction.** Our benchmark focuses on Click-Through Rate (CTR) prediction in a multi-scenario setting. In general CTR prediction (Guo et al., 2017), the CTR value $\hat{y}$ is predicted by a model $\mathcal{F}\theta$, which takes input features $\boldsymbol{x}$ (e.g., user, item, and context features). This is expressed as $\hat{y} = \mathcal{F}_\theta(\boldsymbol{x})$. However, in multi-scenario settings, the input features differ due to the inclusion of scenario-specific features $\boldsymbol{x}_s$ and a scenario indicator $s \in 1, ..., S$, which indicates the scenario to which the input belongs. Additionally, when designing a multi-scenario model $\mathcal{F}_{\theta^M}$, both scenario-specific and shared features must be jointly

considered within the parameter $\theta^M$ across all $S$ scenarios. Mathematically, this is formulated as:

$$\hat{y} = \mathcal{F}_{\theta^M}(\boldsymbol{x}_g, \boldsymbol{x}_s, s), s \in \{1, ..., S\}, \tag{1}$$

here, $\boldsymbol{x}_g$ denotes the general (scenario-independent) features, $\boldsymbol{x}_s$ represents the scenario-specific features for each scenario $s$, and $\hat{y}$ refers to the CTR prediction.

- **Open Datasets.** Open datasets are crucial for research in recommender systems. While numerous datasets are available, their inconsistent usage across studies hinders fair comparisons. Our proposed benchmark addresses this by offering a unified data loading interface, enabling standardized access to datasets. Specifically, we provide several open datasets which have been tested and evaluated under our benchmark. This interface is also designed for easy extensibility, encouraging the use of additional datasets for experimentation and evaluation (see Section 4.1).

- **General Data Processing Methods.** Variations in data processing methods across studies lead to inconsistent results. Most studies use custom methods without sharing processed data or detailed procedures, hindering data reuse. Therefore, our work tries to establish a reproducible data processing paradigm for multiple scenarios, ensuring fair comparison and repeatable experiments. We apply unified processing methods, such as scenario feature declaration and common feature filtering, allowing the community to conduct diverse research with standardized data processing.

- **Unified Model Interface.** Open-source models can be obtained through authors' publications or reproductions by others. However, code package and implementation inconsistencies lead to model output variations. Our benchmark implements standardized modules with a consistent model setup and call interface, ensuring reproducible model implementations and fair performance comparisons through simple hyper-parameter settings. We have implemented ten cutting-edge models for multi-scenario recommendations tested on six commonly used datasets and one industrial dataset, demonstrating the effectiveness of this unified interface.

- **Training.** We have implemented a unified model training procedure to ensure fair comparisons and scalability. This procedure standardizes the training process, allowing for easy extension with various models and datasets. We also provide functions for saving logs, enabling clear record-keeping of training specifics and facilitating the reproducibility of experiments.

- **Evaluation.** Evaluation metrics are critical for assessing model performance. The use of different metrics across studies complicates fair comparisons. To address this, following previous works (Sheng et al., 2021; Yang et al., 2022; Li et al., 2023b; Wang et al., 2023; Chang et al., 2023), we use AUC and Logloss, the two most common metrics, to evaluate model performance across different scenarios. We also provide a consistent evaluation interface for all models, ensuring fair comparisons.

- **Savable Logs & Settings & Tutorial.** We provide a unified interface for hyper-parameter settings to standardize the evaluation process and ensure reproducibility. These settings, along with training logs, are saved in files. This allows users to understand model performance changes during training and easily reproduce results based on the saved settings. Additionally, to facilitate ease of use for researchers, we provide a detailed tutorial that includes environment setup, dataset download, preprocessing, model training, and evaluation. Furthermore, an introduction to manually designed MSR models and datasets is provided to support users in personalized model design.

## 4 BENCHMARKING FOR MULTI-SCENARIO RECOMMENDATIONS

This section offers a concise overview of the datasets used in our benchmark, along with a description of the multi-scenario baseline models we implemented. To highlight our contribution, Table 1 presents a comparison between our benchmark and other well-known recommendation benchmarks. Compared to these, ours is the first benchmark focused on the MSR task and features the most extensive datasets, baseline models, and evaluation pipelines. Furthermore, a more detailed description of the datasets, models, and scenario settings is provided in Appendix A.

### 4.1 DATASET

Adhering to the principles of fair comparison and ease of use, our benchmark selects widely-used multi-scenario open datasets varying in feature numbers and data volumes, Specifically, for public datasets, we choose MovieLens, KuaiRand, Ali-CCP, Amazon, Douban and Mind. Moreover, we

Table 1: Comparison with existing recommender system benchmarks.

| Benchmark | #Models | #MSR Models | #Datasets | #MSR Datasets | Release | MSR Prediction |
|---|---|---|---|---|---|---|
| Spotlight (Kula, 2017) | 8 | 0 | 5 | 0 | 2017 | ✗ |
| DeepCTR (Shen, 2017) | 29 | 4 | 4 | 0 | 2017 | ✗ |
| RecBole (Zhao et al., 2021) | 91 | 0 | 43 | 0 | 2021 | ✗ |
| FuxiCTR Zhu et al. (2021b) | 54 | 5 | 24 | 0 | 2021 | ✗ |
| RecBole-CDR (Zhao et al., 2022) | 10 | 0 | 3 | 0 | 2022 | ✗ |
| SELFRec (Yu et al., 2023) | 16 | 0 | 4 | 0 | 2023 | ✗ |
| **Scenario-Wise Rec** | **12** | **12** | **6** | **6** | **2024** | ✓ |

also provide an industrial dataset from collected from one of the biggest advertising platform to validate these models and the detailed analysis can be found in Section 6. The introduction of the public datasets is elaborated as follows and the dataset statistics are listed in Table 2. We provide more detailed description of datasets in Appendix A.1 and scenarios analysis in Appendix A.3.

Table 2: Dataset statistics for each scenario. † indicates that only a subset of scenarios is presented, see Section 6 for further details.

| Scenario Index | Movie-Lens | | | KuaiRand | | | | | Mind | | | |
|---|---|---|---|---|---|---|---|---|---|---|---|---|
| | S-0 | S-1 | S-2 | S-0 | S-1 | S-2 | S-3 | S-4 | S-0 | S-1 | S-2 | S-3 |
| # Interaction | 210,747 | 395,556 | 393,906 | 2,407,352 | 7,760,237 | 895,385 | 402,366 | 183,403 | 26,057,579 | 11,206,494 | 10,237,589 | 9,226,382 |
| # User | 1,325 | 2,096 | 2,619 | 961 | 991 | 171 | 832 | 832 | 737,687 | 678,268 | 696,918 | 656,970 |
| # Item | 3,429 | 3,508 | 3,595 | 1,596,491 | 2,741,383 | 332,210 | 547,908 | 43,106 | 8,086 | 1,797 | 8,284 | 1,804 |

| Scenario Index | Douban | | | Ali-CCP | | | Amazon | | | Industrial† | | |
|---|---|---|---|---|---|---|---|---|---|---|---|---|
| | S-0 | S-1 | S-2 | S-0 | S-1 | S-2 | S-0 | S-1 | S-2 | S-0 | S-1 | S-2 |
| # Interaction | 227,251 | 179,847 | 1,278,401 | 32,236,951 | 639,897 | 52,439,671 | 198,502 | 278,677 | 346,355 | 301,654 | 91,468 | 22,986 |
| # User | 2,212 | 1,820 | 2,712 | 89,283 | 2,561 | 150,471 | 22,363 | 39,387 | 38,609 | - | - | - |
| # Item | 95,872 | 79,878 | 34,893 | 465,870 | 188,610 | 467,122 | 12,101 | 23,033 | 18,534 | - | - | - |

- **MovieLens** (Harper & Konstan, 2015): The MovieLens[2] dataset contains 1 million ratings for 4 thousand movies by 6 thousand users. It includes user ratings, demographics, and movie metadata. In Scenario-Wise Rec, we divide interaction samples into three age-based scenarios: "1-24", "25-34", and "35+".

- **KuaiRand** (Gao et al., 2022): KuaiRand[3] is an unbiased dataset with 11 million interactions from 1 thousand users and 4 million videos on the Kuaishou App. Scenarios are based on advertising positions, with the top five scenarios used for evaluation.

- **Ali-CCP** (Ma et al., 2018b): Ali-CCP[4] is a large-scale CTR dataset from Taobao's traffic logs. The "301" context feature indicates different scenarios.

- **Amazon** (Haque et al., 2018): The Amazon 5-core dataset[5] is a multi-scenario dataset generated from Amazon. In this paper, three scenarios "Clothing", "Beauty", and "Health" are used for training and evaluation.

- **Douban** (Zhu et al., 2020): The Douban dataset[6] includes subsets for books, music, and movies, with shared users across subsets. Each platform is treated as a distinct scenario, with attributes like "living place" and "user ID" retained.

- **MIND** (Wu et al., 2020): The Microsoft News Dataset (MIND)[7] dataset is for news recommendations. We use training and validation metadata, categorizing the four largest genres ("news", "lifestyle", "sports" and "finance") as separate scenarios.

## 4.2 MULTI-SCENARIO RECOMMENDATION MODEL

With the rapid development of multi-scenario recommendations, research in this field has proliferated. However, variations in data, parameters, and model implementations across studies hinder

---

[2]https://grouplens.org/datasets/movielens/

[3]https://kuairand.com/

[4]https://tianchi.aliyun.com/dataset/408

[5]https://jmcauley.ucsd.edu/data/amazon/

[6]https://www.kaggle.com/datasets/fengzhujoey/douban-datasetratingreviewside-information

[7]https://msnews.github.io/

fair comparisons. To address this, Scenario-Wise Rec reproduces twelve state-of-the-art models frequently mentioned in related studies and evaluates them on six public datasets and one industrial dataset. The following sections introduce these models. Additionally, a detailed introduction to these models is provided in Appendix A.2, and the reproduction details are presented in Appendix B.2

- **Shared Bottom** (Caruana, 1997): The Shared Bottom model uses a shared network to learn common representations for different tasks and applies separate towers for task-specific modeling. It is commonly used in multi-scenario recommendations by treating different scenarios as distinct tasks (Sheng et al., 2021; Wang et al., 2022).

- **MMoE** (Ma et al., 2018a): The Multi-gate Mixture-of-Experts (MMoE) model uses multiple expert networks and gating networks to control connections between experts and task-specific networks. This model is effectively applied in multi-scenario recommendations by treating different scenarios as tasks.

- **PLE** (Tang et al., 2020): The Progressive Layered Extraction (PLE) model mitigates negative transfer and handles complex task correlations in multi-task learning. PLE is particularly effective in multi-scenario recommendations, as it separates shared and task-specific components while employing a progressive routing mechanism.

- **STAR** (Sheng et al., 2021): The Star Topology Adaptive Recommender (STAR) model integrates a shared network for common features and scenario-specific networks tailored to each scenario. This approach enhances both CTR and RPM in Alibaba's advertising system by learning shared and scenario-specific parameters.

- **SAR-Net** (Shen et al., 2021): The Scenario-Aware Ranking Network (SAR-Net) by Alibaba leverages specific attention modules for scenario, item, and user behavior features. It handles biased logs through scenario-specific expert networks and a multi-scenario gating module, demonstrating effectiveness in multi-scenario recommendations.

- **M2M** (Zhang et al., 2022): The Multi-Scenario Multi-Task Meta-Learning (M2M) model captures inter-scenario correlations using a meta unit and meta attention module. It enhances scenario-specific feature representation and is effective for multi-scenario CTR prediction.

- **AdaSparse** (Yang et al., 2022): AdaSparse adapts to scenario-specific sparse structures for multi-scenario CTR prediction by utilizing a lightweight network as a pruner to eliminate redundant information. It demonstrates significant improvements on both public datasets and within Alibaba's advertising system.

- **ADL** (Li et al., 2023a): The Adaptive Distribution Learning Framework (ADL) focuses on multi-scenario CTR prediction with a hierarchical structure that includes clustering and classification. It captures commonalities and distinctions among distributions, demonstrating effectiveness in both public and industrial datasets.

- **EPNet & PPNet** (Chang et al., 2023): PPNet and EPNet, part of the Parameter and Embedding Personalized Network (PEPNet), handle multi-task recommendations under multi-scenario settings. EPNet fuses features with different importance for users, while PPNet modifies parameters for different tasks. These models explore the impact of personalized modifications in multi-scenario recommendations.

- **HAMUR** (Li et al., 2023b): The Hyper Adapter for Multi-Domain Recommendation (HAMUR) is proposed to introduce adapters (Rebuffi et al., 2017) for multi-domain recommendation (MSR) tasks. The adapters are domain-specific, while a shared hyper-network captures domain commonalities dynamically across different domains.

- **M$^3$oE** (Zhang et al., 2024): M$^3$oE introduces a framework consisting of three Mixture-of-Experts (MoE) modules to learn common, domain-specific, and task-specific attributes, along with a two-level fusion mechanism that enables precise control over feature extraction and fusion across different domains and tasks.

## 5 EXPERIMENT

This section presents the results of the benchmark experiment, which includes four main parts: experimental setup, model performance, efficiency analysis, and scenario number analysis, as described below.

Table 3: Performance comparison. The best results are in **bold**. The next best results are underlined. $\pm$ "*" indicates statistical significance (i.e. two-sided t-test with $p < 0.05$).

| Model | Movie-Lens | | KuaiRand | | Ali-CCP | |
|---|---|---|---|---|---|---|
| | AUC↑ | Logloss↓ | AUC↑ | Logloss↓ | AUC↑ | Logloss↓ |
| SharedBottom | 0.8095 ±0.0018 | 0.5228 ±0.0016 | 0.7793 ±0.0009 | 0.5483 ±0.0010 | 0.6232 ±0.0021 | 0.1628 ±0.0012 |
| MMoE | 0.8086 ±0.0020 | 0.5218 ±0.0016 | 0.7794 ±0.0011 | 0.5477 ±0.0012 | 0.6242 ±0.0016 | 0.1621 ±0.0011 |
| PLE | 0.8091 ±0.0013 | 0.5257 ±0.0014 | 0.7796 ±0.0010 | 0.5495 ±0.0010 | 0.6250 ±0.0014 | 0.1617 ±0.0013 |
| STAR | 0.8096 ±0.0015 | 0.5258 ±0.0010 | 0.7806 ±0.0008 | 0.5404 ±0.0010 | 0.6253 ±0.0015 | 0.1613 ±0.0010 |
| SAR-Net | 0.8092 ±0.0014 | 0.5245 ±0.0010 | 0.7816 ±0.0010 | **0.5393**\* ±0.0010 | 0.6245 ±0.0016 | 0.1616 ±0.0010 |
| M2M | 0.8115 ±0.0011 | 0.5213 ±0.0013 | **0.7821**\* ±0.0012 | 0.5397 ±0.0010 | **0.6257**\* ±0.0014 | **0.1611**\* ±0.0011 |
| AdaSparse | 0.8108 ±0.0010 | 0.5205 ±0.0010 | 0.7816 ±0.0011 | 0.5399 ±0.0010 | 0.6239 ±0.0020 | 0.1614 ±0.0012 |
| ADL | 0.8083 ±0.0010 | 0.5238 ±0.0010 | 0.7773 ±0.0008 | 0.5436 ±0.0009 | 0.6233 ±0.0015 | 0.1619 ±0.0012 |
| EPNet | 0.8097 ±0.0019 | 0.5215 ±0.0010 | 0.7801 ±0.0015 | 0.5411 ±0.0013 | 0.6236 ±0.0014 | 0.1612 ±0.0010 |
| PPNet | 0.8063 ±0.0012 | 0.5257 ±0.0012 | 0.7800 ±0.0016 | 0.5408 ±0.0017 | 0.6144 ±0.0009 | 0.1622 ±0.0011 |
| HAMUR | **0.8133**\* ±0.0009 | **0.5193**\* ±0.0011 | 0.7820 ±0.0015 | 0.5397 ±0.0013 | 0.6235 ±0.0011 | 0.1614 ±0.0010 |
| M$^3$oE | 0.8116 ±0.0010 | 0.5211 ±0.0008 | 0.7812 ±0.0011 | 0.5399 ±0.0012 | 0.6249 ±0.0009 | 0.161 ±0.0010 |

| Model | Amazon | | Douban | | Mind | |
|---|---|---|---|---|---|---|
| | AUC↑ | Logloss↓ | AUC↑ | Logloss↓ | AUC↑ | Logloss↓ |
| SharedBottom | 0.6792 ±0.0027 | 0.4790 ±0.0026 | 0.7993 ±0.0011 | 0.5178 ±0.0013 | 0.7509 ±0.0011 | 0.1600 ±0.0014 |
| MMOE | 0.6744 ±0.0025 | 0.4963 ±0.0025 | 0.7978 ±0.0014 | 0.5192 ±0.0010 | 0.7508 ±0.0012 | 0.1600 ±0.0012 |
| PLE | 0.6721 ±0.0020 | 0.4945 ±0.0020 | 0.7977 ±0.0015 | 0.5196 ±0.0017 | 0.7503 ±0.0020 | 0.1601 ±0.0017 |
| STAR | 0.6738 ±0.0022 | 0.4966 ±0.0018 | 0.7957 ±0.0015 | 0.5218 ±0.0017 | **0.7512**\* ±0.0018 | **0.1593**\* ±0.0015 |
| SAR-Net | 0.7071 ±0.0026 | **0.4595**\* ±0.0022 | 0.8033 ±0.0014 | **0.5131**\* ±0.0018 | 0.7490 ±0.0013 | 0.1604 ±0.0015 |
| M2M | 0.6865 ±0.0023 | 0.4943 ±0.0021 | 0.7962 ±0.0014 | 0.5229 ±0.0019 | 0.7508 ±0.0013 | 0.1601 ±0.0017 |
| AdaSparse | 0.6888 ±0.0020 | 0.4831 ±0.0020 | 0.7963 ±0.0013 | 0.5216 ±0.0011 | 0.7497 ±0.0010 | 0.1604 ±0.0019 |
| ADL | 0.7085 ±0.0030 | 0.4658 ±0.0022 | 0.8003 ±0.0012 | 0.5187 ±0.0013 | 0.7328 ±0.0015 | 0.1629 ±0.0021 |
| EPNet | **0.7101**\* ±0.0025 | 0.4688 ±0.0024 | 0.7997 ±0.0014 | 0.5182 ±0.0010 | 0.7418 ±0.0017 | 0.1616 ±0.0018 |
| PPNet | 0.6791 ±0.0025 | 0.4730 ±0.0022 | 0.7994 ±0.0010 | 0.5175 ±0.0009 | 0.7494 ±0.0018 | 0.1603 ±0.0014 |
| HAMUR | 0.6730 ±0.0022 | 0.4890 ±0.0019 | 0.7979 ±0.0012 | 0.5197 ±0.0011 | 0.7494 ±0.0015 | 0.1603 ±0.0015 |
| M$^3$oE | 0.7010 ±0.0019 | 0.4698 ±0.0018 | **0.8036**\* ±0.0010 | 0.5140 ±0.0009 | 0.7451 ±0.0012 | 0.1612 ±0.0011 |

## 5.1 BENCHMARKING SETTINGS

We evaluated twelve models across six public datasets and open-sourced our benchmark package. For datasets, we independently process features for each dataset using discretization and bucketing methods. Features are categorized into three groups: sparse features (discretized attributes), dense features (continuous attributes), and scenario features (scenario-specific operations). The datasets are split into training, evaluation, and testing sets in an 8:1:1 ratio for most datasets. Instead, Ali-CCP is pre-divided into three folds. For evaluation metrics, we follow methodologies from prior MSR works like (Sheng et al., 2021; Yang et al., 2022; Li et al., 2023b; Wang et al., 2023; Chang et al., 2023), using Area Under the ROC Curve (AUC) and Logloss as metrics. AUC measures the probability that a random positive sample ranks higher than a negative one, while Logloss evaluates classification performance. Higher AUC or lower Logloss indicates better model performance. For parameter settings, we ensure a fair comparison by configuring each model within a consistent search space and maintaining similar parameter magnitudes across datasets. All experiments are run 10 times with different random seeds to ensure the robustness of the results. More detailed reproduction information, including parameter settings and model reproduction, can be found in Appendix B.2.

### 5.1.1 COMPREHENSIVE ANALYSIS

The overall results are presented in Table 3, with dataset-specific results shown in Table 9 to 14.

In the experiments, we highlight the challenge of managing the "seesaw effect" through effective scenario correlation modeling. The critical factor is the model's ability to handle varying data distributions across scenarios, avoiding overfitting in data-rich environments while preserving performance in data-sparse ones. This underscores the importance of fine-grained modeling of scenario relationships in multi-scenario approaches.

In Table 3, models leveraging an expert structure (E.g., MMoE, PLE, SAR-Net, M$^3$oE) commonly outperform models that directly model different scenarios (E.g., SharedBottom, ADL), suggesting the former's superior capability in capturing complex inter-scenario dynamics at deeper network lev-

Table 4: Efficiency analysis. "Training" denotes the average training time per epoch and the "Inference" denotes inference time per batch on the test set, batch size is 9,048 for KuaiRand, 102,400 for Ali-CCP and 4,096 for the rest.

| Model | MovieLens | | | Ali-CCP | | | Amazon | | |
|---|---|---|---|---|---|---|---|---|---|
| | Training(s) | Inference(ms) | Param Size | Training(s) | Inference(ms) | Param Size | Training(s) | Inference(ms) | Param Size |
| SharedBottom | 8.68 | 5.49 | 227.59K | 2918.22 | 29.20 | 25.69M | 3.09 | 3.61 | 2.22M |
| MMoE | 9.89 | 5.16 | 217.80K | 3100.01 | 26.50 | 25.40M | 4.49 | 4.15 | 2.21M |
| PLE | 8.17 | 6.16 | 224.20K | 2559.67 | 29.37 | 25.96M | 5.57 | 4.25 | 2.22M |
| STAR | 8.72 | 4.88 | 308.63K | 2992.08 | 30.99 | 25.54M | 5.87 | 4.60 | 2.27M |
| SAR-Net | 7.05 | 7.64 | 239.34K | 2880.83 | 29.77 | 25.07M | 4.06 | 3.95 | 2.23M |
| M2M | 11.71 | 11.83 | 372.53K | 3042.11 | 28.09 | 26.68M | 13.59 | 11.71 | 2.31M |
| AdaSparse | 8.11 | 4.02 | 230.32K | 2885.73 | 27.70 | 25.33M | 3.70 | 3.80 | 2.22M |
| ADL | 8.54 | 4.18 | 257.49K | 3194.35 | 28.69 | 25.52M | 5.86 | 4.49 | 2.24M |
| EPNet | 8.65 | 4.29 | 232.33K | 3014.37 | 29.45 | 25.23M | 4.76 | 3.98 | 2.22M |
| PPNet | 9.83 | 4.32 | 349.68K | 2910.49 | 27.11 | 26.23M | 4.38 | 4.12 | 2.36M |
| HAMUR | 9.88 | 6.96 | 362.43K | 3015.65 | 29.23 | 27.62M | 5.21 | 4.28 | 2.38M |
| $M^3oE$ | 8.92 | 5.85 | 296.57K | 2996.32 | 30.02 | 25.65M | 4.95 | 4.05 | 2.27M |

| Model | Douban | | | KuaiRand | | | Mind | | |
|---|---|---|---|---|---|---|---|---|---|
| | Training(s) | Inference(ms) | Param Size | Training(s) | Inference(ms) | Param Size | Training(s) | Inference(ms) | Param Size |
| SharedBottom | 9.83 | 3.18 | 3.43M | 372.54 | 6.80 | 69.53M | 440.18 | 6.38 | 12.35M |
| MMoE | 11.06 | 2.99 | 3.42M | 398.51 | 8.63 | 69.51M | 449.05 | 6.67 | 12.31M |
| PLE | 11.42 | 3.77 | 3.43M | 370.02 | 9.46 | 69.81M | 537.14 | 8.62 | 12.35M |
| STAR | 11.23 | 4.63 | 3.50M | 355.32 | 9.21 | 69.90M | 448.23 | 8.14 | 12.38M |
| SAR-Net | 10.08 | 4.08 | 3.44M | 330.12 | 6.76 | 69.59M | 410.71 | 6.52 | 12.31M |
| M2M | 18.02 | 9.01 | 3.54M | 357.25 | 13.83 | 72.87M | 553.64 | 11.71 | 12.38M |
| AdaSparse | 10.23 | 2.53 | 3.43M | 331.01 | 5.79 | 69.79M | 471.53 | 4.38 | 12.34M |
| ADL | 10.36 | 2.64 | 3.45M | 358.30 | 4.83 | 69.56M | 439.51 | 4.08 | 12.44M |
| EPNet | 10.03 | 3.02 | 3.43M | 360.04 | 4.64 | 69.95M | 450.68 | 4.33 | 12.30M |
| PPNet | 12.04 | 4.21 | 3.60M | 380.04 | 5.31 | 70.54M | 525.83 | 4.42 | 12.52M |
| HAMUR | 14.29 | 7.68 | 3.77M | 368.32 | 7.65 | 71.32M | 523.56 | 7.81 | 12.36M |
| $M^3oE$ | 13.56 | 6.32 | 3.42M | 364.25 | 6.98 | 69.36M | 478.63 | 6.85 | 12.21M |

els. Furthermore, Models that could dynamically adjust major structures or parameters (E.g., M2M, AdaSparse, HAMUR) depending on different scenarios surpass those with static expert structures, indicating a more precise control over hidden structures' influence on scenario performance. This leads to enhanced scenario correlation understanding and overall model performance. Besides, we could also summarize that dataset size does not directly correlate with model performance disparity.

Additionally, we observe that variability in sparse scenario performance significantly affects overall model effectiveness. Top-performing models maintain high performance across all scenarios, while less effective models show improvements only in specific sparse scenarios. For example, in Ali-CCP, as shown in Table 11, models like STAR and M2M leverage collaborative shared towers and meta-learning to balance domains, enhancing performance in sparse scenario S-1 without compromising performance in dense scenarios S-0 and S-2. This results in superior overall performance, emphasizing the importance of modeling scenario correlations to mitigate the impact of scenario-specific sparsity and facilitate stable performance improvements across all scenarios.

### 5.1.2 EFFICIENCY ANALYSIS

In evaluating efficiency, we present the results, including training time, evaluation time, and parameter size for each model across different datasets, as shown in Table 4, for reference. Adhering to the principles of a fair comparison, we observed that models exhibited a range of parameter sizes, which highlighted the trade-offs between model complexity and efficiency. For relatively small datasets, such as MovieLens and Douban, the training times were notably lower, reflecting the reduced computational load compared to larger dataset Ali-CCP. It is evident that model efficiency is influenced not only by algorithmic design but also significantly by the characteristics of the dataset, including the number and intrinsic nature of features. This is a crucial consideration for applications with limited computational resources. Across different models, the model sizes remained within the same order of magnitude, primarily because most parameters in recommender systems derive from embedding parameters. Our findings underscore the importance of selecting the appropriate model based on both the computational budget and the dataset's specific characteristics. We believe these efficiency results could serve as an essential reference for scholars to select suitable models or datasets based on their resources in practical machine learning applications.

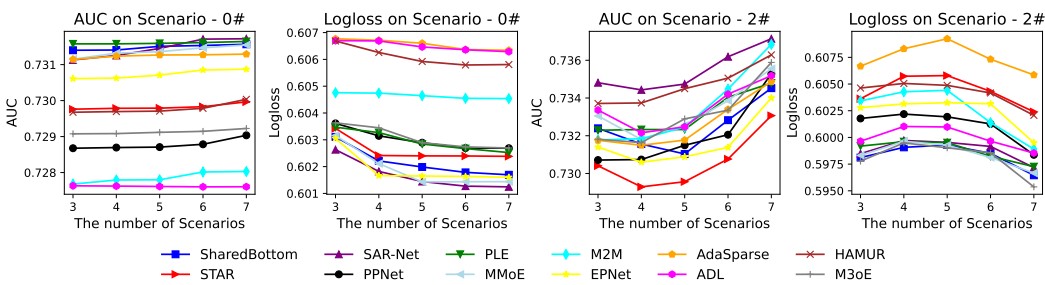

Figure 3: Scenario number analysis on Scenario-0# and Scenario-2#.

### 5.1.3 SCENARIO NUMBER ANALYSIS

In MSR systems, a complex relationship exists between the number of scenarios and the performance of each scenario. In this section, we analyze this relationship using the KuaiRand dataset by varying the number of scenarios from 3 to 7 and observing the resulting performance changes in each model. The experimental settings are detailed in Appendix B.3.

We report the results from two selected scenarios: a dense scenario (Scenario-0#) and a sparse scenario (Scenario-2#). As shown in Figure 3, the performance of both scenarios improves as the number of scenarios increases from 3 to 7. This improvement can be attributed to the increased number of instances, which augments the dataset and enhances domain collaboration, thus boosting overall performance. However, in sparse Scenario-2#, we observe a "seesaw effect", where an initial performance drop is followed by an improvement. This drop is due to the addition of the sparse scenario negatively affecting overall performance, as observed in models like SharedBottom, ADL, and SATR. Notably, SAR-Net demonstrates a strong ability to balance performance across both dense and sparse scenarios, maintaining consistent results. In practical deployments, it is essential to balance the trade-off between performance fluctuations across multiple scenarios and adapt the model to specific conditions.

## 6 INDUSTRIAL EXPERIMENT

The multi-scenario recommendation task is highly relevant to real-world recommendation systems. Compared to public datasets and online recommendation systems, online multi-scenario settings can be more complex due to the greater number and diversity of scenarios, as well as the inclusion of a wider range of features, which current public datasets cannot provide. Therefore, to (1) validate the feasibility of our benchmark in practical settings and (2) provide a reliable benchmark for industrial applications, we tested our benchmark using an industrial dataset[8] from one online tech company's advertising platform. This dataset includes 10 different scenarios and 108 features, spanning nine days. The first seven days are used for training, while the last two are reserved for validation and testing. It covers both common and scenario-specific user and item spaces. Details about the dataset can be found in Table 5.

### 6.1 RESULT ANALYSIS

Table 6 presents the results on the industrial dataset. Compared to other datasets, this industrial dataset features a significantly larger number of scenarios, allowing us to explore how scenario count impacts performance metrics. It is observed that M2M, ADL, and M$^3$oE exhibit superior performance, demonstrating their ability to handle multiple scenarios jointly. This is attributed to their innovative designs, including the meta cell, dynamic routing mechanism, and multi-level fusion mechanism, aligning with the analysis in Section 5.1.1. More scenario-specific results and analysis are provided in Appendix C.7.

---

[8]We will release this dataset upon acceptance to foster research on MSR.

Table 5: Industrial dataset reference sheet.

| Number of Features | 108 |
|---|---|
| Number of Scenarios | 10 |
| Interaction | 3M |
| Features Categories | 1. User features: attributes related to the user's profile and behavior, such as user city, click history, etc. 2. App features: attributes related to the specific application or service being used, such as application category, application size, etc. 3. Context features: context features that users interact with, such as device name, time, domain id, etc. |
| Train/Val/Test Splitting | 7:1:1 (Split by days) |
| Scenario Interaction | S-0: 301,654; S-1: 91,468; S-2: 22,986; S-3: 10,928; S-4: 316,734; S-5: 16,288; S-6: 383,791; S-7: 459,370; S-8: 87,353; S-9: 655,569 |

Table 6: Performance comparison on the industrial dataset.

| Metric/Model | SharedBottom | MMoE | PLE | STAR | SAR-Net | M2M | AdaSparse | ADL | EPNet | PPNet | HAMUR | M$^3$oE |
|---|---|---|---|---|---|---|---|---|---|---|---|---|
| AUC | 0.8276 | 0.8301 | 0.8330 | 0.8310 | 0.8355 | **0.8392** | 0.8224 | 0.8358 | 0.8349 | 0.8318 | 0.8353 | 0.8384 |
| Logloss | 0.1521 | 0.1567 | 0.1496 | 0.1503 | 0.1528 | 0.1494 | 0.1596 | **0.1489** | 0.1517 | 0.1555 | 0.1501 | 0.1492 |

## 6.2 ETHICAL CLARIFICATION

For the industrial dataset, we provide a comprehensive cheatsheet that allows users to quickly review the key aspects of the dataset. The results are presented in Table 5. During the dataset's utilization, ethical considerations are given primary importance during the dataset's utilization, as outlined below:

- **Data Privacy:** (1) Strong measures are implemented to protect sensitive user information. Specifically, user-specific identifiers, such as user IDs, are removed to prevent any risk of sensitive data leakage. (2) Demographic attributes, including gender, province, and city, are transformed into numerical features through a rehashing process, ensuring that the data cannot be reverse-engineered. (3) Behavioral data is similarly anonymized and hashed into numerical values, with explicit user consent obtained prior to data collection. (4) Moreover, the dataset only includes explicit user interactions, such as clicks, while features like favorites, likes, and comments are excluded.

- **Consent:** The data collection process adheres strictly to all relevant legal and regulatory requirements. All data is gathered from a single online platform with user authorization and signed consent. No data is collected from users who have not provided explicit consent.

## 7 CONCLUSION

In this paper, we introduce Scenario-Wise Rec, a pioneering benchmark designed specifically to tackle the complexities and challenges inherent in MSR systems. Scenario-Wise Rec aims to establish a comprehensive framework for facilitating fair and reproducible comparisons among diverse multi-scenario recommendation models, while also promoting the sharing of insights and advancements within this field. Our contributions are threefold. Firstly, to the best of our knowledge, Scenario-Wise Rec is the first benchmark released in the field of multi-scenario recommendation, offering significant benefits for the community by enabling fair comparisons across different models and fostering development. Secondly, we have integrated a pipeline that includes multi-scenario data processing, training, evaluation, along with logging and open-source practices. Scenario-Wise Rec thus sets a new standard for transparency and reproducibility in the field and is friendly for all scholars. Thirdly, we provide the reproduction for twelve multi-scenario recommendation models and seven distinct multi-scenario datasets (six public datasets and one industrial dataset), offering scholars diverse angles to test and implement their models in this field. This facilitates a deeper understanding of the current landscape and identifies potential avenues for future research. We hope our benchmark will contribute to the field and collectively foster collaboration in the area of MSR.

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

## A DATASET AND MODEL DESCRIPTIONS

In this section, a detailed description of the datasets employed in our benchmark is provided, along with an in-depth analysis of scenario-specific information and a description of the multi-scenario baseline models that we implemented in this benchmark.

### A.1 DATASET

Adhering to the principles of fair comparison and ease of use, our benchmark selects six widely-used multi-scenario open datasets varying in feature numbers and data volumes. Furthermore, the benchmark model is deployed on a real-world dataset from an advertising platform to augment the reliability and applicability of experimental comparisons. Specifically, for public datasets, we choose MovieLens-1M, KuaiRand, Mind, Douban, Ali-CCP and Amazon, and the industrial advertising dataset is derived from daily logs. A detailed introduction of these datasets is elaborated as follows:

- **MovieLens** (Harper & Konstan, 2015): The MovieLens dataset is a comprehensive collection of movie ratings and information that is widely used for various research and recommender systems. It contains user ratings, demographic information, movie metadata, and user preferences. It consists of 1 million anonymous ratings of approximately 4 thousand movies made by 6 thousand MovieLens users. With the development of recommender systems, it has become an invaluable resource that enables insights into movie preferences and aids in the development of innovative recommendation systems for the benefit of movie enthusiasts worldwide. In the proposed Scenario-Wise Rec, to realize multi-scenario evaluation, interaction samples are divided into three scenarios based on the "age" feature, i.e., "1-24", "25-34", and "35+".

- **KuaiRand** (Gao et al., 2022): The KuaiRand dataset is an unbiased recommendation dataset with randomly exposed videos gathered from the Kuaishou App. In Scenario-Wise Rec, KuaiRand has been processed and used for model evaluation. It contains 11 million interactions with 1 thousand users and 4 million videos. In this dataset, different scenarios represent different advertising positions of the Kuaishou App. The scenario identification "tab" has already been given as a feature in the range of [0,14] to indicate the scenario of different interactions. To facilitate the evaluation, we extracted data from the top five scenarios with the most data for training and testing.

- **Ali-CCP** (Ma et al., 2018b): Ali-CCP is a large-scale CTR recommendation dataset gathered from the real-world traffic logs of the recommender system in Taobao, which is one of the largest online retail platforms in the world. In this dataset, context feature "301" is regarded as a different scenarios indicator, representing an expression of the position the interaction sample is from.

- **Amazon** (Cui et al., 2020): The Amazon 5-core dataset is a multi-scenario dataset generated from Amazon. In this paper, three scenarios "Clothing", "Beauty", and "Health" are used for training and evaluation.

- **Douban** (Zhu et al., 2020): The Douban dataset, a real-world collection derived from the Douban platform, is divided into three subsets: Douban-book, Douban-music, and Douban-movie. All subsets share the same users, and we treat each platform as a distinct scenario. In terms of user features, attributes like "living place" and "user ID" are retained. For items, we systematically renumber all items across the three scenarios and assign new ids. Following the previous work (Zhu et al., 2020), ratings above 3 are considered positive, while those 3 or below are deemed negative.

- **Mind** (Wu et al., 2020): The MIcrosoft News Dataset (MIND) is specifically designed for news recommendation by Microsoft. It is a real-world dataset gathered from users of the Microsoft News platform. For our benchmark, we collect the metadata from both training and validation datasets of MIND to create a comprehensive dataset. Regarding item features, we maintain "category" and "subcategory" attributes, labeling "clicks" as positive and "not click" as negative. In terms of scenario division, we categorize different genres as separate scenarios. Specifically, we retain the four largest genres, "news", "lifestyle", "sports", and "finance" as distinct scenarios. This configuration encompasses a total of 748 million users, more than 20k items, and over 56 million interactions.

- **Industrial Dataset**: The industrial dataset utilized in our paper is a subset, uniformly sampled from the click logs across ten scenarios on an advertising platform, spanning a nine-day period.

We set the initial seven days' data for training, and the data from the eighth and ninth serve as validation and test datasets, respectively. This dataset comprises 108 features, encompassing user features, item features, contextual features, and scenario-specific features. While different scenarios exhibit a common user and item space, they also maintain their unique scenario-specific users and items.

## A.2 MULTI-SCENARIO RECOMMENDATION MODEL

With the rapid development of multi-scenario recommendations, more and more research has arisen. However, due to the different data, parameters, and model implementation methods used in different studies, it is difficult to directly summarize the current frontier research and make a fair comparison. In order to track the most cutting-edge research in the field of multi-scenario recommendation and facilitate fair comparison, in the proposed Scenario-Wise Rec, we reproduce twelve cutting-edge models that are commonly used or mentioned in the related studies and evaluate them on the six public datasets. We reproduce these models under the uniform model interface, and reproduction details are depicted in Appendix B.2. An introduction about these models is described as follows.

- **Shared Bottom** (Caruana, 1997): The Shared Bottom model is an approach for multi-task recommendation tasks. It learns a shared representation from different tasks with a shared network base to capture the patterns and shared information. Afterward, different network towers are applied to different tasks for task-specific modeling. Recently, it has also been applied to multi-scenario recommendations as a commonly used baseline by treating different scenarios as different recommendation tasks (Sheng et al., 2021; Wang et al., 2022).

- **MMoE** (Ma et al., 2018a): Multi-gate Mixture-of-Experts (MMoE) model is a commonly used model for multi-task learning. Different from the Shared Bottom, MMoE applies multiple expert networks named MOE (i.e., Mixture-of-Experts structure) as the bottom structure and uses multiple gating networks to control the connections between different experts and the following task-specific networks. Through a detailed modeling of task relations, MMoE achieves better performance in multi-task recommendations. Similar to other multi-task models. MMoE can also be easily applied to multi-scenario recommendations by treating different scenarios as different recommendation tasks.

- **PLE** (Tang et al., 2020): The Progressive Layered Extraction (PLE) model is a solution to the challenges faced by multi-task learning (MTL) models in recommender systems. PLE addresses the issues of negative transfer and complex task correlations by separating shared components and task-specific components explicitly and adopting a progressive routing mechanism to gradually extract deeper semantic knowledge. Through extensive experiments, PLE has outperformed state-of-the-art MTL models significantly in various task correlation scenarios. Similarly, PLE could also be applied as an effective multi-scenario recommendation model by treating different scenarios as different recommendation tasks.

- **STAR** (Sheng et al., 2021): The Star Topology Adaptive Recommender (STAR) model addresses the challenge of making click-through rate (CTR) predictions for multiple scenarios within a large-scale commercial platform. It achieves multi-scenario learning by combining a shared network that captures commonalities between scenarios with scenario-specific networks tailored to each scenario. The weights of the shared network and the scenario-specific network are multiplied to generate a unified network during the inference stage for each scenario. STAR effectively learns the shared network from all data and adapts scenario-specific parameters to each scenario's characteristics. Production data has validated the effectiveness of STAR, with significant improvements in CTR and Revenue Per Mille (RPM) observed since its deployment in Alibaba's display advertising system in late 2020.

- **SAR-Net** (Shen et al., 2021): The Scenario-Aware Ranking Network (SAR-Net) is proposed by Alibaba and designed for the travel marketing platform for multi-scenario recommendation tasks. It tackles the challenge of training a unified model by leveraging specific attention modules that incorporate scenario, item features, and user behavior features. Moreover, SAR-Net handles biased logs resulting from manual intervention during promotion periods through scenario-specific expert networks, scenario-shared expert networks, and a multi-scenario gating module. Experiments and online A/B testing demonstrate the effectiveness of SAR-Net, which has been successfully deployed and serves hundreds of travel scenarios on Alibaba's online travel marketing platform.

- **M2M** (Zhang et al., 2022): The Multi-Scenario Multi-Task Meta-Learning (M2M) model is a novel approach designed to address the challenges of multi-task and multi-scenario advertiser modeling in e-commerce platforms like Taobao and Amazon. M2M utilizes a meta unit to capture inter-scenario correlations, a meta attention module to capture diverse inter-scenario correlations for different tasks, and a meta tower module to enhance scenario-specific feature representation for different recommendation tasks. In Scenario-Wise Rec, the number of the meta-towers is set to 1 to correspond to the single CTR prediction task.

- **AdaSparse** (Yang et al., 2022): AdaSparse is designed for multi-scenario CTR prediction and aims to adaptively learn the sparse structures of scenario models. Specifically, AdaSparse introduces a lightweight network functioning as a pruner, which operates a scenario-pruning process for each layer within individual scenario towers. During this pruning process, a novel fusion strategy is employed, combining binary and scale approaches to enhance pruning performance, effectively eliminating as much redundant information as possible. The results demonstrate significant improvements not only in public datasets but also in online A/B tests within Alibaba's advertising system's CTR platform.

- **ADL** (Li et al., 2023a): The Adaptive Distribution Learning Framework (ADL), a novel multi-distribution method, concentrates on multi-scenario CTR prediction. It features an end-to-end, hierarchical structure that includes a clustering process and a classification process. The core component, the distribution adaptation module, employs a routing mechanism, adaptively determining the distribution cluster for each sample. This model effectively captures the commonalities and distinctions among various distributions, thereby enhancing the model's representation capability without relying on prior knowledge for predefined data allocation. Extensive experiments are conducted on public datasets, and an industrial dataset from Alibaba's online system consisting of 10 distinct scenarios. The results demonstrate its effectiveness and efficiency compared to other models.

- **EPNet & PPNet** (Chang et al., 2023): PPNet and EPNet are two submodels in the Parameter and Embedding Personalized Network (PEPNet). EPNet performs personalized selection on embedding to fuse features with different importance for different users in multiple scenarios. PPNet executes personalized modification on DNN parameters to balance targets with different sparsity for different users in multiple tasks. By applying PPNet and EPNet, PEPNet is able to handle multi-task recommendations under multi-scenario settings. In Scenario-Wise Rec, We designed these two models to explore the impact of each on multi-scenario recommendations. Meanwhile, the number of the meta-towers in PPNet is set to the same as the scenario number to correspond to the CTR prediction task on each scenario.

- **HAMUR** (Li et al., 2023b): The HAMUR (Hyper Adapter for Multi-Domain Recommendation) comprises two main components: a domain-specific adapter and a domain-shared hyper-network. The domain-specific adapter is a modular component that can be seamlessly integrated into various recommendation models, allowing each domain to maintain unique adaptations. The domain-shared hyper-network dynamically generates parameters for these adapters by implicitly capturing shared patterns among domains. HAMUR's dynamic architecture is validated through experiments multiple public datasets, demonstrating its ability to outperform state-of-the-art models by enhancing predictive accuracy across diverse domains.

- **M$^3$oE** (Zhang et al., 2024): The M$^3$oE framework, introduced as the Multi-Domain Multi-Task Mixture-of-Experts recommendation system, is designed to tackle complex recommendation challenges across diverse domains and tasks. At its core, M3oE employs three distinct mixture-of-experts (MoE) modules, each dedicated to managing domain preferences and task-specific behaviors. Furthermore, it integrates a two-level fusion mechanism to effectively combine features across both domains and tasks. The framework's adaptability is enhanced through the use of AutoML, which dynamically optimizes its structure, enabling efficient cross-domain and cross-task knowledge transfer, ultimately demonstrating superior performance.

### A.3 SCENARIO INFORMATION ANALYSIS

As mentioned in the previous section, our study employs six public and one industrial datasets. However, unlike conventional recommendation benchmarks, our research primarily targets multi-scenario recommendation tasks. Accordingly, this section provides a detailed analysis of each dataset's scenario-specific information and statistical data.

### A.3.1 SCENARIO SPLITTING STRATEGY

Unlike traditional CTR prediction tasks, MSR models emphasize scenario-unified prediction, requiring a scenario indicator within the dataset features to facilitate dataset splitting. Traditionally, scholars utilize features such as the advertising area, product page number, or other manually defined context features as scenario indicators. Specifically, for datasets focusing on multi-scenario recommendations (E.g., Ali-CCP, KuaiRand), the scenario indicator is often a predefined feature field provided by the dataset itself, representing different sources of different samples (E.g., different advertising slots). For general datasets (E.g., ML-1M), when applied to multi-scenario recommendations, existing studies often use a feature that can clearly distinguish samples as a scenario indicator (E.g., item category). Notably, recent studies, like (Guo et al., 2023), have begun exploring other scenario-splitting features to enhance overall performance. In our benchmark, to advance scenario analysis, we implement various splitting strategies, encompassing traditional context feature division, user feature separation, and item feature segmentation across five datasets. As an example, for the Ali-CCP dataset, we follow the approach of previous studies such as (Wang et al., 2022; Li et al., 2023b), employing the "301" feature, which denotes the display position of items on the screen. In the KuaiRand dataset, segmentation is based on the "tab" feature, indicating whether the recommendation appears on the app's main page or a specific recommendation page. The scenario splitting methods of other datasets are also illustrated in Section A.1.

Table 7: Dataset statistics for scenario intersection.

| Dataset | COV | Scenario Indicator | # User Intersection | # Item Intersection |
|---|---|---|---|---|
| MovieLens | 0.3186 | S-0 ∩ S-1 | - | 3,320 |
| | | S-1 ∩ S-2 | - | 3,448 |
| | | S-0 ∩ S-2 | - | 3,354 |
| KuaiRand | 1.3552 | S-0 ∩ S-1 | 961 | 380,375 |
| | | S-0 ∩ S-2 | 160 | 64,292 |
| | | S-1 ∩ S-2 | 162 | 213,106 |
| | | S-1 ∩ S-3 | 832 | 264,931 |
| | | S-2 ∩ S-3 | 141 | 66,063 |
| | | S-3 ∩ S-4 | 704 | 2,721 |
| Ali-CCP | 0.9180 | S-0 ∩ S-1 | 814 | 188,510 |
| | | S-1 ∩ S-2 | 515 | 188,590 |
| | | S-0 ∩ S-2 | 2,385 | 465,694 |
| Amazon | 0.2696 | S-0 ∩ S-1 | 4,220 | - |
| | | S-1 ∩ S-2 | 6,557 | - |
| | | S-0 ∩ S-2 | 7,026 | - |
| Douban | 1.1053 | S-0 ∩ S-1 | 1,736 | - |
| | | S-1 ∩ S-2 | 1,815 | - |
| | | S-0 ∩ S-2 | 2,209 | - |
| Mind | 0.5611 | S-0 ∩ S-1 | 675,343 | - |
| | | S-1 ∩ S-2 | 646,049 | - |
| | | S-2 ∩ S-3 | 633,042 | - |
| | | S-0 ∩ S-2 | 689,568 | - |
| | | S-1 ∩ S-3 | 626,604 | - |
| | | S-0 ∩ S-3 | 653,595 | - |

### A.3.2 SCENARIO ANALYSIS

The results of the dataset splitting are detailed in Table 2 of the original paper. Considering the variability in splitting outcomes across different datasets, we utilize the Coefficient of Variation (COV) (Everitt & Skrondal, 2010) to evaluate the uniformity of scenario distribution within each dataset. A higher COV value signifies a higher degree of uneven distribution among scenarios, as depicted in Table 7. Our analysis indicates that KuaiRand exhibits the most uneven scenario distribution, and MovieLens displays the most uniform distribution. This observation aligns with our splitting strategy. MovieLens is segmented into relatively evenly distributed age groups. In contrast, KuaiRand users tend to mainly stay on the homepage, leading to an uneven distribution across different pages. The Douban dataset is uneven because the browsing history for movies is greater than that for books and music. The COV values for the Ali-CCP datasets are approximately 0.9, indicating a non-uniform distribution across all scenarios. In contrast, the Mind and Amazon datasets exhibit a more even distribution across different scenarios, as evidenced by their lower COV values.

To gain a deeper understanding of scenario splitting in public datasets, we illustrate the intersection of different scenarios in each dataset in Table 7. However, for the industrial dataset, owing to data protection and privacy policies, obtaining specific user and item information is not feasible. Our findings indicate that user and item interaction attributes vary significantly across different datasets. In the MovieLens dataset, segmented by users' age groups, we observe that each age group shares a majority of movies while maintaining a distinct preference for a small number of films. For KuaiRand, we notice a bimodal distribution in scenario users and a long-tail distribution in items. This pattern is also reflected in interaction distribution. For example, scenarios 3 and 4 share 704 users out of a total of 832, suggesting similar user behavior patterns in these scenarios, yet the interactions with items are notably distinct. In the Ali-CCP dataset, Scenario 1 is quite small, accounting for nearly 1% of total interactions, resulting in a skewed scenario distribution. Intersection analysis reveals that these three scenarios maintain distinct attributes, sharing only a small portion of users and items across each pair. In Amazon, Douban, and Mind datasets, since these three datasets do not have Scenario-specific features, thus we take different splitting strategies. For Amazon datasets, different scenarios represent different items intersection for different scenarios in the Amazon platform, thus we find that they share a large number of users, but the interactions across different scenarios are evenly distributed. For Douban, scenarios are split by different platforms, including "Book", "Music" and "Movie". The movie has the most browsing histories, but these three platforms share over 1,000 users. The same for the Mind dataset, we split scenarios by different news categories, as users browse different news feeds on the platform, they share the most users, over 600,000.

## B EXPERIMENT SETTINGS

### B.1 IMPLEMENTATION DETAILS

In this part, we present the experiment setting during our experiment. Our framework is implemented using PyTorch. Empirically, we set the feature embedding dimension $d$ to 16. We customized batch sizes for each dataset: 4096 for MovieLens, Amazon, Douban and Mind, 9,048 for both Kuairand and the industrial dataset, and 102,400 for Aliccp. Experiments were conducted on a single GPU of Tesla V100 PCIe 32GB, utilizing the Adam optimizer. The initial learning rate was set to 1e-3. To enhance training performance, we incorporated an early stopping strategy and a learning rate scheduler for optimal adjustment. All experiments were conducted three times under different random seeds.

### B.2 MODEL REPRODUCTION DETAILS

In this part, we provide the reproduction details for each model, serving as a reference for users.

- **SharedBottom**: Our SharedBottom code implementation comprises a single-layer MLP at the bottom, followed by scenario-specific MLP towers for each scenario. Considering the dataset sizes, we configured the MLP towers with three layers for the MovieLens, KuaiRand, Douban, Mind, and Industrial datasets and six layers for dataset Aliccp. We search the dimension bottom layer in {128, 256, 512}.

- **MMoE**: Our MMoE module is consistent with the original paper (Ma et al., 2018a). During our experiment, we search the space of expert dimension {128, 256, 512} and for the output tower, without loss of generality, we choose six layers MLP for Aliccp and other datasets for three layers of MLP.

- **PLE**: In our PLE implementation, unlike the implementation used in multi-task recommendation models, we replaced the task-specific and task-shared experts with scenario-specific and scenario-shared experts. Our exploration space including CGC layers {1, 2} and expert dimensions {128, 256, 512}. Regarding the output tower design, we adhered to the configurations employed in both MMoE and Shared Bottom models.

- **STAR**: In reproducing the STAR model, our implementation remains strictly consistent with the specifications outlined in the original paper. We employ a single-layer network for the auxiliary network, and for the scenario tower, MLPs are utilized. The configuration of the scenario tower is set with three layers for all the datasets except for Aliccp, aligning with previous settings. We explored auxiliary network dimensions within the searching space {8, 16, 32}.

- **SAR-Net**: In SAR-Net implementation, there are deviations from the method described in the original paper. Specifically, we omitted the cross-scenario behavior extraction layer, a design intended to process user behavior sequences, because our datasets lack such features. Consequently, this module was excluded from our implementation. Our exploration space for the configuration included scenario-shared expert counts within $\{2, 4, 8\}$ and scenario-specific expert counts within $\{1, 2\}$.

- **M2M**: In our reproduction of the M2M model, which originally focus on multi-scenario multi-task problems, our work focuses on a single task—CTR prediction. Thus, accordingly, we adapted it for a single-task tower. Our exploration space comprised the expert output size within $\{8, 16\}$, the number of encoding layers within $\{1, 2\}$, the number of decoding layers within $\{2, 3\}$, and the feedforward dimension within $\{128, 256, 512\}$.

- **AdaSparse**: In our replication of the AdaSparse model, as detailed in the original paper (Yang et al., 2022), we initially employ a scenario-adaptive pruner module. This module offers three instantiation strategies: "Binarization", "Scaling", and "Fusion". Each represents distinct approaches to computing weighting factors. Subsequently, this adaptive pruning technique is utilized to facilitate a sparse MLP for CTR prediction across varied scenarios, demonstrating its flexibility in handling sparse data environments. We employ the "Fusion" strategy for all datasets, without losing generality. The backbone network is chosen for three and six, respectively, for different datasets like Aliccp and Kuairand. And we set $\alpha$ to 1 and the searching space for $\beta$ is $\{2, 3, 4\}$.

- **ADL**: In the reproduction of the ADL model, we commence by establishing a shared fully connected network dedicated to modeling correlations across different scenarios. This is complemented by the construction of several scenario-specific fully connected networks, aimed at conducting nuanced, scenario-specific analyses. Furthermore, a Distribution Learning Module (DLM) is developed as illustrated in the original paper (Li et al., 2023a) , employing a clustering algorithm based on cosine similarities to enable dynamic routing during both training and inference phases, thereby enhancing the model's adaptability to diverse data distributions. For the shared fully-connected network, we follow the previously mentioned setting: three layers for dataset Movie-Lens, KuaiRand, Amazon, Douban, Mind, industrial dataset, and six layers for Aliccp. Besides, we search the space of the number of clusters in $\{3, 4, 5\}$.

- **EPNet**: In constructing the EPNet, we first built the Gate NU module to provide gated scaling signals for the model. Then, we divide the input into scenario-side features and scenario-agnostic features (i.e., sparse features and dense features), respectively, and embed them into embedding vectors. Afterward, we construct the scaled embedding by inputting the scenario-side embedding and detached scenario-agnostic embeddings to the GateNU module and applying the output scaling parameters to the original embedding. To avoid the effects of the PPNet structure, through a simple parameter search, we replace the subsequent network about PPNet in the original paper with a three-layer or six-layers feedforward structure with different neurons according to different datasets and add an output header to output values between [0, 1].

- **PPNet**: In developing the PPNet model, we adhered to the design outlined in paper (Chang et al., 2023). Initially, we concatenate ID embeddings and input them into Gate NU modules. The number of Gate NU modules is the same as the number of PPNet layers. Subsequently, we constructed the PPNet tower. Given that PPNet was originally designed for multi-task learning, we adhered to our initial settings, assigning different task-specific architectures within the scenario tower. We configured PPNet with MLPs tailored to various dataset distributions to adhere to the settings like previous models, six-layer MLP for Aliccp, and three-layers for the others. For each instance, the input is directed to an appropriate scenario tower based on its "scenario indicator".

- **HAMUR**: In developing HAMUR Li et al. (2023b), we followed the settings outlined in the original paper. We selected the feature domain ID as the domain indicator, and for different datasets, different model architectures were chosen. For the Ali-CCP dataset, a seven-layer MLP was selected as the backbone model, while for the other datasets, only a three-layer MLP was used. Regarding the hyper-network, a single-layer MLP with a hidden dimension of 64 was set, but different hyper-matrix sizes were used. For the seven-layer backbone model, the hyper-matrix size was set to 65, while for the others, it was set to 35.

- **M$^3$oE**: In reproducing M$^3$oE (Zhang et al., 2024), we follow the original paper but made a modification by setting the task number to 1, making it compatible with multi-scenario prediction. We used the sparse and dense features along with the domain ID as domain indicators. For the parameters, we set the number of experts within the search space of $\{3, 4, 5\}$ across all datasets.

Regarding the dimension setting, we specified a five-layer model for prediction, in accordance with the reproduction instructions outlined in the original paper.

### B.3 Scenario Number Experiment Details

During the scenario number experiment, we selected the Kuairand dataset for various numbers of scenarios. This choice is due to the fixed number of scenarios in other datasets like Ali-CCP, Douban, etc., whereas the Kuairand dataset allows for the selection of different numbers of scenario subsets by specifying the feature "tab". To validate the effect of the number of scenarios, we selected the top-3 to top-7 scenarios from the original Kuairand dataset (e.g., 3 scenarios correspond to scenarios 0-2). The statistics are recorded in Table 8. For the principle of fair comparison, we set all model hyper-parameters to be the same, specifically, we configured "tower_params", "mlp_params", and "fcn_dims" for different models as a two-layer MLP with dimensions [64,32].

Table 8: Scenario distribution for scenario-number experiments.

| Scenario | # Interaction |
|---|---|
| Scenario 0 | 7,760,237 |
| Scenario 1 | 2,407,352 |
| Scenario 2 | 895,385 |
| Scenario 3 | 402,366 |
| Scenario 4 | 183,403 |
| Scenario 5 | 37,418 |
| Scenario 6 | 17,430 |

## C Experimental Analysis on Different Datasets

This section provide a detailed experimental analysis on the different datasets based on Table 3, which is the same as Table 3 in the original paper.

### C.1 Analysis for Movie-Lens

As Table 2 demonstrates, the distribution of all scenarios in the MovieLens dataset is quite balanced. Analyzing the overall performance from Table 3, HAMUR, M2M and AdaSparse emerge as the top performance models. This success is attributed to the design of the dynamic metrix ,meta unit and the sparse pruner, which effectively recognizes scenario-specific patterns, allowing the model to adapt across all scenarios. Table 9 reveals no significant "seesaw phenomenon", aligning with our dataset splitting strategy. However, structural differences among models result in varied scenario emphases. For instance, Shared-Bottom models, which share a bottom tower across all scenarios, exhibit a more uniform performance than other MSR models.

### C.2 Analysis for KuaiRand

KuaiRand is a dataset comprising five distinct scenarios, which, unlike the MovieLens dataset, shows an uneven distribution across scenarios. Analysis of Table 3 reveals that MSR models such as SAR-Net, HAMUR, and M2M significantly outperform multi-task models like SharedBottom, MMoE, and PLE. This underscores the importance of meticulous architecture design for multi-scenario tasks, considering that variations in data distribution across different scenarios can have a profound impact on overall performance. The "seesaw phenomenon" observed in Table 10 illustrates the disparity in performance across scenarios, with scenarios 2# and 4# significantly outperforming the others.

### C.3 Analysis for Ali-CCP

Ali-CCP is a dataset containing three scenarios, with a notably uneven distribution due to the small size of scenario 1#. Analysis of Table 3 indicates that STAR and M2M lead other models by a

Table 9: The scenario-detailed results for Movie-Lens. The best results are in **bold**. The next best results are underlined.

| Models/AUC | Total | S-0 | S-1 | S-2 |
|---|---|---|---|---|
| Shared Bottom | 0.8095 | 0.8116 | 0.8128 | 0.8041 |
| MMoE | 0.8086 | 0.8029 | 0.8178 | 0.8016 |
| PLE | 0.8091 | 0.8118 | 0.8186 | 0.8002 |
| STAR | 0.8096 | 0.8137 | 0.8133 | 0.7979 |
| SAR-Net | 0.8092 | 0.8068 | 0.8158 | 0.8026 |
| M2M | 0.8115 | 0.8111 | 0.8163 | **0.8057** |
| AdaSparse | 0.8108 | 0.8109 | **0.8188** | 0.7947 |
| ADL | 0.8083 | 0.8074 | 0.8160 | 0.7995 |
| EPNet | 0.8097 | 0.8100 | 0.8148 | 0.8031 |
| PPNet | 0.8063 | 0.8084 | 0.8113 | 0.7994 |
| HAMUR | **0.8133** | **0.8160** | 0.8186 | 0.8056 |
| M$^3$oE | 0.8116 | 0.8127 | 0.8169 | 0.8050 |

| Models/Logloss | Total | S-0 | S-1 | S-2 |
|---|---|---|---|---|
| Shared Bottom | 0.5228 | 0.5243 | 0.5208 | 0.5239 |
| MMoE | 0.5218 | 0.5239 | 0.5164 | 0.5262 |
| PLE | 0.5257 | 0.5335 | 0.5164 | 0.5310 |
| STAR | 0.5258 | 0.5239 | 0.5228 | 0.5299 |
| SAR-Net | 0.5245 | 0.5337 | 0.5180 | 0.5261 |
| M2M | 0.5213 | 0.5321 | 0.5208 | 0.5240 |
| AdaSparse | 0.5205 | 0.5248 | 0.5137 | 0.5400 |
| ADL | 0.5238 | 0.5293 | 0.5162 | 0.5283 |
| EPNet | 0.5215 | 0.5251 | 0.5178 | 0.5234 |
| PPNet | 0.5257 | 0.5266 | 0.5228 | 0.5281 |
| HAMUR | **0.5180** | **0.5206** | **0.5131** | **0.5215** |
| M$^3$oE | 0.5211 | 0.5259 | 0.5171 | 0.5224 |

Table 10: The scenario-detailed results for KuaiRand. The best results are in **bold**. The next best results are underlined.

| Models/AUC | Total | S-0 | S-1 | S-2 | S-3 | S-4 |
|---|---|---|---|---|---|---|
| Shared Bottom | 0.7793 | 0.7117 | 0.7282 | 0.7898 | 0.7293 | 0.8535 |
| MMoE | 0.7794 | 0.7146 | 0.7272 | 0.7773 | 0.7310 | 0.8562 |
| PLE | 0.7796 | 0.7104 | 0.7285 | 0.7890 | 0.7298 | 0.8531 |
| STAR | 0.7806 | 0.7201 | 0.7305 | 0.7895 | 0.7322 | 0.8055 |
| SAR-Net | 0.7816 | **0.7263** | 0.7312 | **0.7921** | **0.7359** | 0.8378 |
| M2M | **0.7821** | 0.7248 | **0.7326** | 0.7898 | 0.7339 | 0.8447 |
| AdaSparse | 0.7816 | 0.7243 | 0.7314 | 0.7889 | 0.7332 | 0.8227 |
| ADL | 0.7773 | 0.7258 | 0.7244 | 0.7887 | 0.7349 | 0.8071 |
| EPNet | 0.7801 | 0.7235 | 0.7303 | 0.7883 | 0.7319 | 0.7803 |
| PPNet | 0.7800 | 0.7167 | 0.7285 | 0.7887 | 0.7329 | **0.8642** |
| HAMUR | 0.7820 | 0.7225 | 0.7323 | 0.7903 | 0.7340 | 0.8486 |
| M$^3$oE | 0.7812 | 0.7251 | 0.7312 | 0.7918 | 0.7342 | 0.7984 |

| Models/Logloss | Total | S-0 | S-1 | S-2 | S-3 | S-4 |
|---|---|---|---|---|---|---|
| Shared Bottom | 0.5483 | 0.3532 | 0.6074 | 0.5357 | 0.6092 | 0.3454 |
| MMoE | 0.5477 | 0.3510 | 0.6069 | 0.5507 | 0.6110 | **0.3344** |
| PLE | 0.5495 | 0.3517 | 0.6092 | 0.5479 | 0.6078 | 0.3444 |
| STAR | 0.5404 | 0.3335 | 0.6019 | 0.5331 | 0.6003 | 0.3753 |
| SAR-Net | **0.5393** | **0.3319** | 0.6014 | **0.5307** | 0.6023 | 0.3467 |
| M2M | 0.5397 | 0.3324 | 0.6012 | 0.5340 | 0.6011 | 0.3436 |
| AdaSparse | 0.5399 | 0.3333 | 0.6014 | 0.5350 | 0.6015 | 0.3604 |
| ADL | 0.5436 | 0.3369 | 0.6064 | 0.5330 | **0.5986** | 0.3875 |
| EPNet | 0.5411 | 0.3340 | 0.6022 | 0.5344 | 0.6013 | 0.3942 |
| PPNet | 0.5408 | 0.3353 | 0.6033 | 0.5331 | 0.6006 | 0.3491 |
| HAMUR | 0.5397 | 0.3331 | **0.6004** | 0.5324 | 0.5999 | 0.3351 |
| M$^3$oE | 0.5399 | 0.3325 | 0.6013 | 0.5314 | 0.6010 | 0.3782 |

narrow margin. This suggests that the design of the star topology and the meta-unit paradigm can effectively address balance across all scenarios, especially in cases of significant unevenness in scenario distribution. Regarding scenario-specific results in Table 11, the seesaw effect is evident, particularly since STAR and M2M demonstrate superior performance in the data-sparse scenario 1#, outperforming other models significantly.

Table 11: The scenario-detailed results for Ali-CCP. The best results are in **bold**. The next best results are underlined.

| Models/AUC | Total | S-0 | S-1 | S-2 |
|---|---|---|---|---|
| Shared Bottom | 0.6232 | 0.6279 | 0.5627 | 0.6246 |
| MMoE | 0.6242 | 0.6279 | 0.5744 | 0.6247 |
| PLE | 0.6250 | 0.6280 | 0.5841 | 0.6245 |
| STAR | 0.6253 | 0.6270 | **0.6041** | 0.6242 |
| SAR-Net | 0.6245 | **0.6282** | 0.5900 | **0.6253** |
| M2M | **0.6257** | 0.6278 | 0.6018 | 0.6247 |
| AdaSparse | 0.6239 | 0.6220 | 0.5926 | 0.6237 |
| ADL | 0.6233 | 0.6249 | 0.5823 | 0.6222 |
| EPNet | 0.6236 | 0.6257 | 0.5974 | 0.6222 |
| PPNet | 0.6144 | 0.6156 | 0.5591 | 0.6144 |
| HAMUR | 0.6235 | 0.6258 | 0.5978 | 0.6218 |
| M$^3$oE | 0.6249 | 0.6270 | 0.6021 | 0.6237 |

| Models/Logloss | Total | S-0 | S-1 | S-2 |
|---|---|---|---|---|
| Shared Bottom | 0.1628 | 0.1659 | 0.2001 | 0.1605 |
| MMoE | 0.1621 | 0.1652 | 0.1801 | 0.1600 |
| PLE | 0.1617 | 0.1653 | 0.1810 | 0.1597 |
| STAR | 0.1613 | 0.1650 | 0.1786 | 0.1588 |
| SAR-Net | 0.1616 | **0.1646** | 0.1797 | 0.1589 |
| M2M | **0.1611** | 0.1649 | 0.1788 | **0.1585** |
| AdaSparse | 0.1614 | 0.1660 | 0.1793 | 0.1594 |
| ADL | 0.1619 | 0.1651 | 0.1795 | 0.1587 |
| EPNet | 0.1612 | 0.1648 | 0.1790 | 0.1587 |
| PPNet | 0.1622 | 0.1655 | 0.1881 | 0.1599 |
| HAMUR | 0.1614 | 0.1649 | 0.1786 | 0.1586 |
| M$^3$oE | 0.1616 | **0.1646** | **0.1782** | 0.1587 |

## C.4 ANALYSIS FOR AMAZON

Three scenarios were selected from the original raw datasets of Amazon-5 core: "Beauty", "Clothing", and "Health". Each pair of scenarios shares nearly a thousand users, as indicated in Table 7. The results in Table 12 demonstrate that EPNet and ADL outperform other models. This indicates that gate unit in EPNet and the cluster routing mechanism within ADL effectively capture the commonalities shared by users across different scenarios. Furthermore, conventional multi-task models did not achieve good performance due to their inability to balance the trade-offs among the scenarios.

## C.5 ANALYSIS FOR DOUBAN

The Douban dataset comprises three scenarios: "Book", "Music", and "Movie". The distribution of these scenarios is quite uneven, with scenario 2# having 1,278,401 intersections, significantly more than the other scenarios. As shown in Table 13, scenario 2# dominates the results. Additionally, SAR-Net consistently performs the best across all MSR models, effectively balancing the trade-offs between different scenarios, such as scenario 0# and scenario 2#.

## C.6 ANALYSIS FOR MIND

The Mind dataset was specifically collected for news recommendation. We selected four different scenarios: "news", "lifestyle", "sports" and "finance". The performance results are presented in Table 14. All scenarios share a large number of users, and the distribution of scenarios is comparatively unbalanced, with scenario #0 being the dominant scenario. We found that STAR achieved the best performance, which we attribute to its sharing mechanism. STAR employs a "hard-sharing" method that directly shares an MLP across all scenarios. SharedBottom and MMoE also use the hard-sharing method, resulting in their superior performance. Additionally, we found that M2M achieved great performance, suggesting that the meta-unit can compete effectively with hard-sharing methods.

## C.7 ANALYSIS FOR INDUSTRIAL DATASET

Our industrial dataset, derived from log samples on one of an advertising platforms, encompasses ten distinct scenarios. We present the overall performance results in Table 15. In comparison to

Table 12: The scenario-detailed results for Amazon. The best results are in **bold**. The next best results are underlined.

| Models/AUC | Total | S-0 | S-1 | S-2 |
|---|---|---|---|---|
| Shared Bottom | 0.6792 | 0.6826 | 0.6509 | 0.7026 |
| MMoE | 0.6744 | 0.6730 | 0.6448 | 0.6964 |
| PLE | 0.6721 | 0.6742 | 0.6405 | 0.6983 |
| STAR | 0.6738 | 0.6731 | 0.6444 | 0.6966 |
| SAR-Net | 0.7071 | 0.7069 | 0.6780 | 0.7276 |
| M2M | 0.6865 | 0.6874 | 0.6582 | 0.7083 |
| AdaSparse | 0.6888 | 0.6897 | 0.6618 | 0.7073 |
| ADL | 0.7085 | 0.7083 | 0.6775 | 0.7306 |
| EPNet | **0.7101** | **0.7092** | **0.6794** | **0.7323** |
| PPNet | 0.6791 | 0.6797 | 0.6435 | 0.7031 |
| HAMUR | 0.6730 | 0.6735 | 0.6427 | 0.6971 |
| $M^3oE$ | 0.7010 | 0.7029 | 0.6716 | 0.7235 |

| Models/Logloss | Total | S-0 | S-1 | S-2 |
|---|---|---|---|---|
| Shared Bottom | 0.4790 | 0.5027 | 0.4925 | 0.4546 |
| MMoE | 0.4963 | 0.5219 | 0.5164 | 0.4654 |
| PLE | 0.4945 | 0.5187 | 0.5204 | 0.4598 |
| STAR | 0.4966 | 0.5175 | 0.5198 | 0.4659 |
| SAR-Net | 0.4695 | **0.4832** | **0.4737** | **0.4344** |
| M2M | 0.4943 | 0.5100 | 0.5154 | 0.4683 |
| AdaSparse | 0.4831 | 0.5022 | 0.5018 | 0.4571 |
| ADL | **0.4658** | 0.4892 | 0.4834 | 0.4383 |
| EPNet | 0.4688 | 0.4934 | 0.4874 | 0.4396 |
| PPNet | 0.4730 | 0.4965 | 0.4872 | 0.4480 |
| HAMUR | 0.4890 | 0.5158 | 0.5004 | 0.4643 |
| $M^3oE$ | 0.4698 | 0.4943 | 0.4879 | 0.4412 |

Table 13: The scenario-detailed results for Douban. The best results are in **bold**. The next best results are underlined.

| Models/AUC | Total | S-0 | S-1 | S-2 |
|---|---|---|---|---|
| SharedBottom | 0.7993 | 0.7144 | 0.7349 | 0.8119 |
| MMoE | 0.7978 | 0.7098 | 0.7317 | 0.8111 |
| PLE | 0.7979 | 0.7142 | 0.7342 | 0.8109 |
| STAR | 0.7957 | 0.7080 | 0.7292 | 0.8089 |
| SAR-Net | 0.8033 | **0.7220** | **0.7451** | 0.8154 |
| M2M | 0.7962 | 0.7004 | 0.7160 | 0.8145 |
| AdaSparse | 0.7963 | 0.7073 | 0.7279 | 0.8096 |
| ADL | 0.8003 | 0.7124 | 0.7287 | 0.8142 |
| EPNet | 0.7997 | 0.7129 | 0.7281 | 0.8132 |
| PPNet | 0.7994 | 0.7119 | 0.7384 | 0.8122 |
| HAMUR | 0.7979 | 0.7101 | 0.7373 | 0.8108 |
| $M^3oE$ | **0.8036** | 0.7190 | 0.7399 | **0.8169** |

| Models/Logloss | Total | S-0 | S-1 | S-2 |
|---|---|---|---|---|
| SharedBottom | 0.5178 | 0.5531 | 0.4952 | 0.5147 |
| MMoE | 0.5192 | 0.5563 | 0.4981 | 0.5156 |
| PLE | 0.5196 | 0.5543 | 0.4955 | 0.5169 |
| STAR | 0.5218 | 0.5581 | 0.4998 | 0.5185 |
| SAR-Net | **0.5131** | **0.5487** | **0.4895** | 0.5101 |
| M2M | 0.5229 | 0.5681 | 0.5147 | 0.5160 |
| AdaSparse | 0.5216 | 0.5577 | 0.4997 | 0.5184 |
| ADL | 0.5187 | 0.5604 | 0.5018 | 0.5137 |
| EPNet | 0.5182 | 0.5551 | 0.4986 | 0.5144 |
| PPNet | 0.5175 | 0.5548 | 0.4931 | 0.5143 |
| HAMUR | 0.5197 | 0.5574 | 0.4933 | 0.5167 |
| $M^3oE$ | 0.5140 | 0.5530 | 0.4935 | **0.5099** |

Table 14: The scenario-detailed results for Mind. The best results are in **bold**. The next best results are underlined.

| Models/AUC | Total | S-0 | S-1 | S-2 | S-3 |
|---|---|---|---|---|---|
| SharedBottom | 0.7505 | 0.7675 | 0.6992 | 0.7561 | 0.7336 |
| MMoE | 0.7504 | 0.7670 | 0.7001 | 0.7560 | 0.7338 |
| PLE | 0.7503 | 0.7668 | 0.6993 | 0.7565 | 0.7331 |
| STAR | **0.7512** | **0.7678** | _0.7007_ | **0.7577** | **0.7351** |
| SAR-Net | 0.7490 | 0.7653 | 0.6984 | 0.7557 | 0.7338 |
| M2M | _0.7508_ | _0.7675_ | **0.7010** | _0.7566_ | _0.7344_ |
| AdaSparse | 0.7497 | 0.7664 | 0.6999 | 0.7564 | 0.7341 |
| ADL | 0.7328 | 0.7480 | 0.6737 | 0.7444 | 0.7203 |
| EPNet | 0.7418 | 0.7599 | 0.6806 | 0.7493 | 0.7294 |
| PPNet | 0.7494 | 0.7661 | 0.6992 | 0.7555 | 0.7330 |
| HAMUR | 0.7494 | 0.7655 | 0.7001 | 0.7563 | 0.7334 |
| M$^3$oE | 0.7451 | 0.7624 | 0.6933 | 0.7533 | 0.7282 |

| Models/Logloss | Total | S-0 | S-1 | S-2 | S-3 |
|---|---|---|---|---|---|
| SharedBottom | 0.1600 | 0.1578 | 0.1662 | 0.1823 | 0.1361 |
| MMoE | 0.1616 | 0.1578 | 0.1662 | 0.1830 | 0.1357 |
| PLE | 0.1610 | 0.1579 | 0.1662 | 0.1824 | 0.1362 |
| STAR | **0.1601** | _0.1576_ | 0.1662 | 0.1821 | 0.1357 |
| SAR-Net | 0.1604 | 0.1582 | 0.1666 | _0.1817_ | _0.1354_ |
| M2M | _0.1602_ | **0.1574** | _0.1661_ | **0.1816** | **0.1352** |
| AdaSparse | 0.1644 | 0.1622 | 0.1699 | 0.1854 | 0.1407 |
| ADL | 0.1629 | 0.1611 | 0.1695 | 0.1839 | 0.1368 |
| EPNet | 0.1616 | 0.1593 | 0.1688 | 0.1830 | 0.1358 |
| PPNet | 0.1603 | 0.1580 | 0.1663 | 0.1818 | 0.1355 |
| HAMUR | 0.1603 | 0.1580 | **0.1660** | 0.1821 | 0.1359 |
| M$^3$oE | 0.1612 | 0.1590 | 0.1669 | 0.1826 | 0.1370 |

other datasets, this industrial dataset features a significantly larger number of scenarios, facilitating our investigation into how scenario number influences performance metrics and the observation of the "seesaw phenomenon". It is observed that M$^3$oE, SAR-Net and M2M exhibit superior performance on this dataset, demonstrating their enhanced ability to capture scenario-specific features when faced with a large number of scenarios, attributing to the innovative design of the scenario-specific transformer and meta cell.

Table 15: The scenario-detailed results for Industrial Dataset. The best results are in **bold**. The next best results are underlined.

| Models/AUC | Total | S-0 | S-1 | S-2 | S-3 | S-4 | S-5 | S-6 | S-7 | S-8 | S-9 |
|---|---|---|---|---|---|---|---|---|---|---|---|
| Shared Bottom | 0.8276 | 0.6480 | 0.7176 | 0.8194 | 0.7451 | 0.8238 | 0.8740 | 0.8420 | 0.6833 | 0.7653 | 0.8227 |
| MMoE | 0.8301 | 0.6484 | 0.7251 | 0.8808 | 0.7351 | 0.8251 | 0.8501 | 0.8407 | 0.7241 | _0.7752_ | 0.8371 |
| PLE | 0.8330 | 0.6494 | 0.7240 | 0.8195 | 0.7648 | 0.8195 | _0.9262_ | 0.8474 | 0.6999 | 0.7317 | 0.8323 |
| STAR | 0.8310 | 0.6449 | _0.7351_ | 0.8071 | 0.7179 | 0.7921 | 0.8529 | 0.8191 | 0.6728 | 0.7024 | 0.8109 |
| SAR-Net | 0.8355 | 0.6580 | **0.7382** | _0.8903_ | _0.7678_ | **0.8286** | **0.9598** | 0.8484 | **0.7413** | 0.7581 | 0.8417 |
| M2M | **0.8392** | 0.6534 | 0.7114 | 0.8770 | 0.7584 | 0.8257 | 0.8823 | 0.8504 | _0.7256_ | 0.7596 | _0.8462_ |
| AdaSparse | 0.8354 | 0.6428 | 0.7350 | 0.8821 | 0.7489 | 0.7617 | 0.9122 | 0.8387 | 0.6854 | 0.7629 | 0.8328 |
| ADL | 0.8358 | _0.6592_ | 0.7103 | **0.8969** | 0.7605 | 0.8254 | 0.9219 | **0.8534** | 0.7145 | **0.7808** | 0.8460 |
| EPNet | 0.8349 | 0.6413 | 0.6449 | 0.8239 | 0.7646 | 0.8253 | 0.8778 | 0.8414 | 0.716 | 0.7478 | 0.8376 |
| PPNet | 0.8318 | 0.6473 | 0.6265 | 0.8011 | 0.7245 | _0.8284_ | 0.9254 | 0.8321 | 0.7174 | 0.7454 | 0.8401 |
| HAMUR | 0.8353 | 0.6545 | 0.7065 | 0.8512 | 0.7502 | 0.8259 | 0.9100 | 0.8452 | 0.7163 | 0.7705 | 0.8407 |
| M$^3$oE | _0.8334_ | **0.6632** | 0.7102 | 0.8625 | **0.7679** | 0.8185 | 0.8932 | 0.8492 | 0.7194 | 0.7575 | **0.8473** |

| Models/Logloss | Total | S-0 | S-1 | S-2 | S-3 | S-4 | S-5 | S-6 | S-7 | S-8 | S-9 |
|---|---|---|---|---|---|---|---|---|---|---|---|
| Shared Bottom | 0.1521 | 0.1505 | 0.1863 | 0.0853 | 0.1706 | 0.1259 | 0.0362 | 0.2062 | 0.0584 | 0.1497 | 0.1959 |
| MMoE | 0.1567 | 0.1508 | 0.1801 | 0.0779 | 0.1705 | 0.1263 | 0.0281 | 0.2038 | 0.0562 | 0.1535 | 0.1948 |
| PLE | 0.1496 | 0.1514 | 0.1802 | _0.0753_ | 0.1901 | 0.1231 | 0.0311 | 0.2096 | 0.0593 | 0.1521 | 0.2001 |
| STAR | 0.1503 | 0.1632 | **0.1793** | 0.0977 | 0.2006 | _0.1198_ | 0.0532 | 0.2021 | 0.0719 | 0.1574 | 0.2117 |
| SAR-Net | 0.1528 | 0.1509 | 0.1811 | 0.0817 | 0.1941 | 0.1486 | 0.0335 | 0.2336 | 0.0597 | 0.1672 | 0.2108 |
| M2M | 0.1494 | _0.1442_ | 0.182 | 0.0840 | _0.1687_ | 0.126 | 0.0314 | 0.2009 | 0.059 | 0.1488 | _0.1897_ |
| AdaSparse | 0.1596 | 0.1594 | 0.1867 | 0.0922 | 0.1727 | 0.1508 | 0.0297 | 0.218 | 0.0792 | 0.1642 | 0.1968 |
| ADL | **0.1489** | **0.1438** | 0.1843 | **0.0745** | 0.171 | 0.1253 | 0.0277 | **0.1981** | _0.0551_ | _0.1438_ | **0.1861** |
| EPNet | 0.1517 | 0.1509 | 0.1917 | 0.0842 | 0.1784 | 0.1212 | 0.0297 | _0.1993_ | 0.0617 | 0.1483 | 0.1957 |
| PPNet | 0.1555 | 0.1554 | 0.2011 | 0.1014 | 0.1989 | 0.1227 | **0.0254** | 0.2032 | 0.0672 | 0.1622 | 0.1972 |
| HAMUR | 0.1501 | 0.1486 | _0.1796_ | 0.0812 | 0.2012 | **0.1189** | 0.0498 | 0.2102 | 0.0731 | **0.1385** | 0.2096 |
| M$^3$oE | _0.1492_ | 0.1502 | 0.1842 | 0.0947 | **0.168** | 0.1311 | **0.0246** | 0.2012 | **0.0534** | 0.1493 | 0.1998 |

## D  LIMITATION AND FUTURE RESEARCH

In this section, we will discuss the limitation of our benchmark and current multi-scenario recommendation research, furthermore, we also provide future research topic in this realm.

- **Limitation**: Compared to other tasks in recommendation systems, such as multi-task recommendation, multi-behavior recommendation, and multi-modal recommendation, multi-scenario recommendation is a relatively new yet burgeoning research topic. Currently, most research focuses on multi-scenario collaboration to improve click-through rates, which is the primary focus of our benchmark. In the past three months, scholars have begun to explore other tasks in multiple scenario, including how to segment scenarios (Jia et al., 2024), how to use large language models to align semantics between scenarios (Fu et al., 2023), and how to enhance performance in multiple scenarios through causal inference (Zhu et al., 2024). Since most of this research is still in its infancy and due to factors such as not passing peer review or not publishing code implementation details, we only include some well-recognized SOTA models in this field in our benchmark. However, we will update our benchmarks in a timely manner based on the development of multi-scenario research.

- **Future Research**: For future research, several noteworthy topics merit attention. Firstly, refining the application of Large Language Models for fine-grained scenario alignment is crucial. While Uni-CTR (Fu et al., 2023) offers a foundational approach, it does not explicitly extract scenario commonalities, thereby constraining scenario expansion. Secondly, although current Multi-Scenario Recommendation research predominantly focuses on Click-Through Rate (CTR) tasks, other areas such as sequential recommendations for diverse scenarios and trustworthy recommendations within MSR remain underexplored. Finally, developing a joint model that simultaneously considers multiple tasks, scenarios, behaviors, and interests could pave the way for a more generalized recommendation system.

