# OpenReview forum: "Scenario-Wise Rec: A Multi-Scenario Recommendation Benchmark"
_ICLR.cc/2025/Conference — ICLR 2025 Conference Withdrawn Submission_

### Official Review · Reviewer_vgv8 · 2024-10-29

**Soundness:** 3
**Presentation:** 2
**Contribution:** 2
**Rating:** 3
**Confidence:** 4

**Summary:**

This paper presents a benchmark that includes six existing datasets for multi-scenario ctr prediction. Besides, they implement twelve models in these datasets with established data and training pipelines. The datasets and codes are provided.

**Strengths:**

1. The datasets and codes are provided.
2. The topic is meaningful in that researchers can implement baseline comparison via the provided benchmark.
3. The experiments are comprehensive with 10 times running.

**Weaknesses:**

1. Some clarifications are confusing. The terminology requires revision, particularly the use of "recommendation," which does not accurately reflect the concept being discussed.
2. The comparative analysis presented in Table 1 does not sufficiently demonstrate novel contributions relative to existing work in the field.
3. The methodology section would benefit from additional detail, particularly regarding the model parameter selection process and optimization criteria.
4. The motivation and challenge are not convincing. Most of the works in the papers have been provided with runnable code responsity.

**Questions:**

1. Why do authors claim that this research is a cut-edge benchmark, while no related explanations are provided in the paper?
2. It seems that most of the baselines and evaluation metrics are designed for CTR prediction. So why use recommendation expressions in the paper?
3. What is the novel challenge for researchers to implement the multi-scenario models?
4. Can you provide statistical analysis for supporting the second challenge, that many MSR models are closed-sourced.

---

> ### Author Response · Authors · 2024-11-18
> **Feedback to Reviewer vgv8 - 1/2**
>
> Dear Reviewer vgv8,
>
> We truly appreciate your insights. However, there may be some gaps between your understanding and the motivation behind our work. We would be happy to provide clarification to address your concerns.
>
> **W1: Some clarifications are confusing. The terminology requires revision, particularly the use of "recommendation," which does not accurately reflect the concept being discussed.**
>
> **A1**: In this paper, we focus on the field of multi-scenario recommendation. In this area, most existing works primarily focus on CTR prediction. Consequently, the community generally refers to the task of "multi-scenario recommendation" as the task of multi-scenario CTR prediction, unless explicitly stated otherwise. Following this convention, we adopt the same approach in our work.
>
> **W2: The comparative analysis presented in Table 1 does not sufficiently demonstrate novel contributions relative to existing work in the field.**
>
> **A2**: Thank you for raising concerns about our comparison with previous benchmarks. Our benchmark focuses specifically on multi-domain recommendation, a field where, to the best of our knowledge, no prior benchmarks have been established. As illustrated in Table 1, existing benchmarks like Spotlight, DeepCTR, RecBole, FuxiCTR, and SelfRec focus on general recommendation tasks. besides, RecBole-CDR is centered on cross-domain recommendation and None of these works address multi-domain recommendation. To the best of our understanding, our benchmark is the first to focus on this area, encompassing both datasets and models. This is the key distinction we aim to highlight in Table 1, showcasing the comparison between our benchmark and existing ones.
>
>
> **W3: The methodology section would benefit from additional detail, particularly regarding the model parameter selection process and optimization criteria.**
>
> **A3**: We sincerely thank you for raising concerns about model parameter selection and optimization. Due to space constraints, detailed experimental information could not be included in the main content. However, all relevant details, including model implementation and optimization, are provided in Section B of the Appendix. Module design and parameter selection are thoroughly discussed in Section B.2 for each model. Regarding optimization, we utilize a cross-entropy loss function with the Adam optimizer and employ an early stopping strategy to prevent overfitting. These experimental settings are also elaborated in Section B.1 for your reference.
>
>
>
> **W4: The motivation and challenge are not convincing. Most of the works in the papers have been provided with runnable code responsity.**
>
> **A4**: We would like to clarify our contributions by emphasizing that our benchmark is built on an entirely new framework, encompassing data collection, unified reproduction, and evaluation. This goes beyond simply collecting public code repositories to form a benchmark.
>
> * For **SharedBottom, MMoE, and PLE**, the currently available public repositories are designed for multi-task settings. To adapt them to the multi-scenario recommendation setting, we rebuilt the entire framework and reproduced these models accordingly.
>
> * For **STAR, SAR-Net, M2M, AdaSparse, ADL, EPNet, and PPNet**, none of these papers provide public code, as most of them originate from tech companies and are not open-sourced due to commercial considerations. We reproduced these models entirely from scratch.
>
> * Regarding **Hamur and M3oE**, these two works have demonstrated relatively high quality in publishing their code repositories and have received a greater number of citations compared to other papers published around the same time. Therefore, we have used these two papers as key references and considered their implementations as SOTA baselines.
>
> * For other works mentioned in the Related Work section, most of those published before 2023 are not publicly available. After 2023, there has been an increase in publicly available works, including PLATE, M-Scan, UNI-CTR, and Hi-Net, reflecting the community's growth. However, models such as D3, MDRAU, and PEPNet remain unavailable due to commercial restrictions.
>
> Building a unified framework that is both publicly accessible and transparent is crucial for fostering further development in the field, and this forms the core objective of our benchmark.

---

> > ### Comment · Reviewer_vgv8 · 2024-11-20
> >
> > Thank you for your response. What is the difference between multi-domain recommendation and cross-domain recommendation? Can you provide references to support your claim?

---

> > > ### Author Response · Authors · 2024-11-24
> > > **Look forward to your reply**
> > >
> > > Dear Reviewer vgv8,
> > >
> > > We sincerely thank you for your prompt feedback and appreciate your active participation in the discussion. To address your concerns, we have summarized our responses as follows:
> > >
> > > * Explanation of MSR and CTR prediction.
> > > * Further clarification on comparisons with previous benchmarks, including the parameter section and optimization details.
> > > * Highlighting the novel challenges addressed by our benchmark.
> > > * Provision of statistics related to both closed and open-sourced MSR papers.
> > >
> > > We kindly ask if our responses have resolved the questions you raised, and we hope you might consider reevaluating the score of our benchmark. Your insights are invaluable to us, and we welcome any further discussion or feedback.
> > >
> > > Best regards,
> > >
> > > The Authors of Paper 9690

---

> > > > ### Comment · Reviewer_vgv8 · 2024-11-30
> > > >
> > > > Thank you for your detailed response.
> > > >
> > > > Thank you for your careful review of the MSR and CDR components. I have some concerns about the current explanations and would appreciate clarification:
> > > >
> > > > 1. Could you provide the mathematical formulations for both MSR and CDR to better illustrate their key differences? I notice that various papers in the field, such as [1], present different definitions from those given in the current explanation.
> > > >
> > > > 2. There seems to be an inconsistency with the cited reference [2]. The paper defines dual-target and multi-target cross-domain recommendation as focusing on improving performance across multiple domains, which appears to differ from the current explanation.
> > > >
> > > > I believe clarifying these definitions with precise mathematical formulations would help ensure alignment with the existing literature and provide a more accurate theoretical foundation for the work.
> > > >
> > > > Ref:
> > > >
> > > > [1] Sheng, Xiang-Rong, Liqin Zhao, Guorui Zhou, Xinyao Ding, Binding Dai, Qiang Luo, Siran Yang et al. "One model to serve all: Star topology adaptive recommender for multi-domain ctr prediction." In Proceedings of the 30th ACM International Conference on Information & Knowledge Management, pp. 4104-4113. 2021.
> > > >
> > > > [2] Zhu, Feng, et al. "Cross-domain recommendation: challenges, progress, and prospects." arXiv preprint arXiv:2103.01696 (2021).

---

> ### Author Response · Authors · 2024-11-18
> **Feedback to Reviewer vgv8 - 2/2**
>
> **Q1: Why do authors claim that this research is a cut-edge benchmark, while no related explanations are provided in the paper?**
>
>
> **A5**: We apologize for not providing more detailed explanations regarding our statement of "cutting-edge" contributions. In the revised version of the PDF, we have added further explanations in the part of **contribution** to better emphasize our advancements. Please refer to the these paragraphs for these updates.
>
> **Q2: It seems that most of the baselines and evaluation metrics are designed for CTR prediction. So why use recommendation expressions in the paper?**
>
> **A6**: We kindly invite you to refer to A1. In the MSR community, the primary focus is typically on the CTR prediction task. As such, unless specifically noted otherwise, we generally consider the task in multi-scenario recommendation to refer to CTR prediction by default.
>
> **Q3: What is the novel challenge for researchers to implement the multi-scenario models?**
>
> **A7**: Thank you for highlighting your concerns regarding the novel challenges. Currently, there are two significant challenges in the MSR field that our benchmark is designed to address:
>
> - Lack of a unified benchmark: In the MSR field, there is a clear need for a benchmark that offers unified implementations of MSR models along with a pipeline for fair comparisons. Our benchmark is specifically created to fill this gap.
>
> - Model design for various industrial and research needs: As MSR research progresses, especially in industrial applications, the requirements for multi-scenario systems vary widely across platforms. Some scenarios are divided by factors such as location or page number, while others are based on content type or user groups. However, there is currently a lack of benchmarks that offer diverse scenarios and real-world conditions for validation across different use cases. Our benchmark addresses this gap by providing several public datasets featuring various scenario-splitting strategies, along with an industrial dataset for factual validation. Additionally, we include detailed tutorials to guide researchers in designing customized models, fostering the development of more robust and powerful implementations to advance the field. We hope this addresses your concerns.
>
> **Q4: Can you provide statistical analysis for supporting the second challenge, that many MSR models are closed-sourced.**
>
> **A8**: We kindly ask you refer to A4 for our analysis of the current state of open-source research. Additionally, we provide a statistical of all the models mentioned in the related work section, categorizing them as either open-source or closed-source.
>
> |  Type | model  |
> | ------------ | ------------ |
> | open-sourced  | HiNet, HAMUR, PLATE, M-scan, Uni-CTR   |
> | close-sourced  | Shared-bottom (MSR-Version), MMoE(MSR-Version), PLE(MSR-Version), Mario, EPNet, PPNet, STAR, SAML, SAR-Net, ADL, CausalInt, AdaSparse, D3, MDRAU  |
>
> Out of a total of 19 models, only 5 are open-source, with the majority remaining closed-source. Our benchmark incorporates implementations of many of these closed-source models, providing researchers with valuable resources to study and drive further advancements in this field.

---

> ### Author Response · Authors · 2024-11-21
> **Difference between Multi-domain recommendation & Cross-domain recommendation**
>
> We thanks for your prompt response.
>
> Cross-domain recommendation and multi-domain recommendation are two distinct tasks in the field of recommendation systems. In cross-domain recommendation, the focus is typically on two separate domains: one as the source and the other as the target and the primary goal is to transfer knowledge from the source domain to the target domain, one primiary example is cold-start problems, where the target domain contains unseen items or users, necessitating additional knowledge from other domains to supplement it.
>
> In contrast, multi-domain recommendation involves a more flexible knowledge transfer process. Here, domains are treated as both sources and targets, and the objective is not just to learn from a single domain, but to enhance the performance of all domains collectively, thus to enhance the overall performace jointly.
>
> These differences are also discussed in detail in previous works like ADI (Section 2 of the original paper) [1] and PEPNet (Section 4.2 of the original paper) [2] and one survey paper (Section 2.2 of the original paper) [3] discussing the target differences between multi-domain recommendation and cross-domain recommendation. Please let me know if you have any further questions or if anything requires clarification. We are happy to assist further.
>
>
> [1]. Adaptive Domain Interest Network for Multi-domain Recommendation. CIKM'22.
>
> [2]. PEPNet: Parameter and Embedding Personalized Network for Infusing with Personalized Prior Information. KDD'23.
>
> [3]. Cross-Domain Recommendation: Challenges, Progress, and Prospects. IJCAI'21.

---

> ### Author Response · Authors · 2024-12-01
> **Mathematical clarification & Comparison with previous works**
>
> Dear vgv8,
>
> We sincerely thank you for raising your concerns, and we are happy to provide clarification on the issues.
>
> **Q1. Mathematical formulations for MDR and CDR**
>
> Preliminaries:
> * $\boldsymbol{e}$ - Domain features vectors. For example, $\boldsymbol{e}^d$ represents the $d$-th feature of domain $d$.
> * $\mathcal{E}$ - Evaluation function. In our benchmark, this denotes the metrics used to evaluate CTR prediction performance.
> * $\mathcal{f}$ - Transfer function. This function extracts knowledge from one domain and transfers it to another domain.
>
> For **Cross-Domain Recommendation (CDR)**, involving two domain: a source domain $d^s$ and a target domain $d^t$.The objective is to extract transferable knowledge from the source domain to enhance the performance of the target domain. This task can be formally defined as:
> $$
> y^t = \mathcal{E}(\boldsymbol{e}^t,\mathcal{f}_{CDR}(\boldsymbol{e}^s))
> $$
> where:
> - $ y^t $ represents the prediction for the target domain,
> - $ \boldsymbol{e}^t $ denotes feature vector of the target domain,
> - $\mathcal{f}(\boldsymbol{e}^s)$ represents the transformation or mapping of the source domain's feature vector
> $e^s$ to the target domain's feature space via the $\mathcal{f}_{CDR}$. Its main objective it to enhance the performance of target domain.
>
> For **Multi-Domain Recommendation (MDR)**, involving $n$ domain: $d^i, i \in \\{1, \dots,n\\}$. The objective is to extract transferable knowledge from multiple source domain to enhance the performance of all domains collectively. This task can be formally defined as:
> $$
> y^i = \mathcal{E}(\boldsymbol{e}^i, \mathcal{f}_{MDR}(\boldsymbol{e}^j)), \quad i \in \\{1, \dots, n\\}, j \in \\{1, \dots, n\\} \setminus \\{i\\}
> $$
> where
> - $ y^i $ represents the prediction for the domain $d^i$,
> - $\mathcal{f}(\boldsymbol{e}^j)$ represents the transformation or mapping of all the domain features, except for the $i$-th domain feature $\boldsymbol{e}^j$, by the multi-domain model $\mathcal{f}_{MDR}$ to jointly enhance the performance across $n$ domains.
>
> From the above, it is clear that for MDR, the transfer direction is not unidirectional, as in CDR, but rather bidirectional. Additionally, multiple domains are involved, where information from these domains is extracted and jointly enhances the performance of all $n$ domains simultaneously .
>
> **Q2. Explanation of cited paper [1]**
>
> We kindly invite you refer to to the first paragraph of Section 2.3 in [1], where it written as:
> > _Multi-domain learning enables knowledge transfer between domains to improve learning. As such, it contrasts with the domain adaptation (DA) problem, where knowledge transfer is only one way, i.e., from the source domain to the target domain._
>
> where we believe that this statement is consistent with our previous definitions. Here the CDR is considered a Domain Adaptation problem, while multi-domain learning is viewed as knowledge transfer across domains.
>
> **Q3. Explanation of cited paper [2]**
>
> We sincerely thank you for your thoughtful and careful insight into these differences. To avoid any confusion, we would like to provide clear definition distinctions and a corresponding table for our paper with paper [2], to address your concerns.
> | Definition in paper [2] | Definition in our paper  |
> |-|-|
> | MDR | MDR - Explicit Modeling |
> | Multi-Target CDR | MDR - Implicit Modeling |
> | Dual-target CDR | MDR - Implicit Modeling with only two domains (Special case in MDR)|
>
> **We would like to clarify that paper [2] primarily focuses on the direction of information transfer**. For MDR in [2], they define information transfer as singular across all domains (denoted as 'single target' in the original paper). For Multi-target CDR, information transfer is defined as bidirectional (denoted as 'multi-target' in the original paper), while Dual-target CDR is constrained to only two domains.
>
> The definitions proposed in [2] are a bit vague that primary leading to potential confusion between CDR and MDR, few studies have adopted this definition approach.
> **Therefore, in our paper, we adopt the definitions used in recent papers (e.g., ADI, PEPNet, etc.) and make a clear distinction between multi-domain recommendation and cross-domain recommendation.** In our paper. CDR  is defined as focusing only on scenarios with significant transfer relationships between two domains, and MDR focuses multiple domains.  As for domains with clear directional relationships, we categorize them in "Explicit Modeling" branch in MDR, while methods where the relationships between scenarios are unclear are termed "Implicit Modeling" (the approach most models use).
>
> We hope our explanation provides sufficient clarity to address your concerns, and we promise to include mathematical modeling and a comparison with previous works in the revised version of our paper.

---

### Official Review · Reviewer_ESgT · 2024-10-31

**Soundness:** 2
**Presentation:** 3
**Contribution:** 2
**Rating:** 3
**Confidence:** 3

**Summary:**

The paper presents a new benchmark for Multi-Scenario Recommendation. To deal with the problems of lacking standard data process pipline and closed-source models in MSR research, this paper introduces Scenario-Wise Rec, which includes six public datasets and one industrial dataset, twelve benchmark models, and a standardized training and evaluation pipeline to achieve model comparisons.

The contributions are:
It is the first benchmark specifically designed for MSR, a standardized pipeline for data processing, training and evaluation.
Open-Source and Real-World Applicability: Scenario-Wise Rec is publicly available, and is validated by applying it to a complex industrial dataset.

**Strengths:**

This paper developed a standardized benchmark specifically for Multi-Scenario Recommendation tasks. This idea is good and needed in this area. This platform is particularly novel. It provides a comprehensive framework that includes multiple datasets, implementations, and a full evaluation pipeline, aiming to give comparisons across scenarios. The idea of the standard platform and benchmark is quite important. It provides a valuable tool to address the growing need for reliable, standardized MSR evaluations.

The paper is also well-organized and clear in its presentation. It is easy to read and the idea is presented directly. The authors provide much context about the challenges in MSR and the motivation behind their benchmark. The diagrams and tables effectively illustrate the framework’s components and results, which help the researcher who may want to leverage the benchmark.

**Weaknesses:**

The idea of ​​the article is very good, providing a benchmark. However, because it is a benchmark, it must adapt to different needs and also take into account the multi-scenario interaction of MSR, which is a very difficult task. This paper has an ambiguous definition of multi-scenario. The datasets used are segmented but lack true cross-scenario relationships, which limits the ability to share knowledge between scenarios as MSR ideally should. The experiments focus on isolated scenario performance without evaluating transferability across scenarios.

Also, the use of only AUC and Logloss limits insights into cross-scenario performance consistency. What if other researchers's model have different targets and need different evaluation metrics?

Last, the pipeline may lack modularity, restricting researchers from customizing data processing, model structures, or metrics.

**Questions:**

1. The experimental setup appears focused on individual scenario performance. Could you consider expanding the experiments to include cross-scenario evaluations, such as training a model in one scenario and testing in another, to validate transferability? This would provide valuable insights into the models’ adaptability across scenarios.

2. Could you elaborate on how the current data segmentation aligns with a true multi-scenario framework? Specifically, how do you envision the potential for knowledge transfer between segmented scenarios in the current datasets, given that they do not share users or items across contexts?

---

> ### Author Response · Authors · 2024-11-18
> **Feedback to Reviewer ESgT - 1/2**
>
> Dear reviewer ESgT,
>
> We sincerely thanks for your insightful feedback, especially regarding cross-scenario analysis. We appreciate your comments and would like to address your concerns as follows:
>
> **W1: The datasets used are segmented but lack true cross-scenario relationships, which limits the ability to share knowledge between scenarios as MSR ideally should. The experiments focus on isolated scenario performance without evaluating transferability across scenarios.**
>
> **A1**: We kindly invite you to revisit Table 7 in the Appendix, where scenario interactions are provided. Please note that in Multi-scenario Recommendation (MSR), information sharing (users, items, or both) across scenarios is a fundamental prerequisite. This sharing enables MSR models to explicitly or implicitly extract shared information, thereby enhancing the performance of all scenarios collectively.
>
> Additionally, we would like to clarify the distinction between Cross-scenario Recommendation and Multi-scenario Recommendation, as these are fundamentally different tasks. Cross-scenario Recommendation primarily focuses on issues such as cold-start scenarios, emphasizing knowledge transfer from a source scenario to a target scenario. In contrast, Multi-scenario Recommendation centers on scenario collaboration. In this context, source and target scenarios are not fixed; any scenario can function as both a source and a target. The emphasis lies in extracting collaborative signals across scenarios and balancing scenario-specific contributions to jointly enhance performance across all individual scenarios, thereby improving overall click-through rates and increasing revenue under this framework.
>
> Regarding your mention of measuring "transferability" between scenarios, we wish to clarify that in Multi-scenario Recommendation, scenario transferability is implicitly assumed. This is foundational to multi-scenario modeling, as datasets are often collected based on shared contexts, such as the same app, webpage, or user history. These scenarios inherently contribute to items and users, which differs significantly from the explicit measurement of transferability between source and target domains in Cross-scenario Recommendation.
>
>
> **W2: The use of only AUC and Logloss limits insights into cross-scenario performance consistency.**
>
> **A2**: Thank you for highlighting the importance of metrics. Currently, our benchmark focuses on the multi-scenario CTR prediction task. To the best of our knowledge, AUC and Logloss are the standard metrics used across all MSR tasks, with no alternative metrics being widely adopted at this time. However, we will continue to monitor developments in this field and incorporate additional metrics as new tasks or discoveries emerge.
>
>
> **W3: Last, the pipeline may lack modularity, restricting researchers from customizing data processing, model structures, or metrics.**
>
> **A3**: Thank you for raising concerns about reproducibility. One detail you may have missed is that our benchmark is fully modular. Data processing, model structure, log saving, and scenario evaluation are all designed in a modular manner. Additionally, we have provided a detailed tutorial in our repository for users to reference. Please refer to Sections 4 and 5 of our [code repository](https://anonymous.4open.science/r/Scenario-Wise-Rec-05B5/README.md), as well as the data processing script [run_example.py](https://anonymous.4open.science/r/Scenario-Wise-Rec-05B5/scripts/run_example.py) and the custom model script [base_example.py](https://anonymous.4open.science/r/Scenario-Wise-Rec-05B5/scenario_wise_rec/models/multi_domain/base_example.py). These resources include code templates for building customized data processing, model architectures, and result-saving mechanisms based on user definitions. We hope this addresses your concerns.

---

> ### Author Response · Authors · 2024-11-18
> **Feedback to Reviewer ESgT - 2/2**
>
> **Q1: The experimental setup appears focused on individual scenario performance. Could you consider expanding the experiments to include cross-scenario evaluations, such as training a model in one scenario and testing in another, to validate transferability? This would provide valuable insights into the models’ adaptability across scenarios.**
>
> **A4**: Regarding the cross-domain evaluation tasks you mentioned, there are some existing benchmarks, such as [Recbole-CDR](https://github.com/RUCAIBox/RecBole-CDR/tree/main), which we have also compared in Table 1 of our paper. It is important to note that our work focuses on a different task from cross-domain recommendation. We hope this clarifies the distinction.
>
> **Q2: Could you elaborate on how the current data segmentation aligns with a true multi-scenario framework? Specifically, how do you envision the potential for knowledge transfer between segmented scenarios in the current datasets, given that they do not share users or items across contexts?**
>
> **A5**: We appreciate your questions regarding scenario handling and knowledge transfer after dataset segmentation. Below, we provide a detailed explanation of the process in hopes of addressing your concerns.
>
>
> First, the dataset is segmented into different scenarios based on "scenario features" (e.g., ad locations, pages). It is important to note that in multi-scenario datasets, the segmented scenarios often contain shared information (such as shared users, shared items, or both), which serves as a fundamental prerequisite. Zero knowledge sharing among scenarios does not exist, as this is not a realistic situation in real-world data collection for multi-scenario datasets.
>
> Next, the handling of domain-shared and domain-specific information varies across models. Taking STAR as an example, embedding parameters are shared across domains, meaning that the embedding spaces for different scenarios are identical. However, the processing of embeddings differs between scenarios. STAR uses a domain-shared tower and a domain-specific tower to explicitly extract shared and independent information. For information fusion, STAR adopts a dot-product mechanism in a "star-shaped" structure, ensuring effective knowledge transfer across domains.
>
> For details on how other models handle shared information, please refer to Appendix A.2, where we provide comprehensive descriptions. We hope this clarifies your concerns.

---

> > ### Author Response · Authors · 2024-11-24
> > **Look forward to your feedback**
> >
> > Dear Reviewer M76W,
> >
> > We sincerely appreciate your insightful feedback and questions, particularly regarding the connections to cross-scenario recommendations. During the discussion period, we provided clarifications and made modifications to address your concerns, summarized as follows:
> >
> > * Clarification of relations within different scenarios for the current datasets.
> > * Discussion on the rationale behind the metrics used in our study.
> > * Explanation of the modular design of our benchmark.
> > * Clarification of the differences between cross-scenario recommendation and multi-scenario recommendation.
> > * Explanation of knowledge transfer within multi-scenario recommendation under our framework.
> >
> > We would greatly appreciate it if you could let us know whether our responses have addressed your concerns and kindly consider reevaluating our paper. We are eager to hear any further feedback you might have and are looking forward to your valuable insights!
> >
> > Best regards,
> >
> > The Authors of Paper 9690

---

> > ### Author Response · Authors · 2024-12-01
> > **Look forward to your feedback**
> >
> > Dear Reviewer ESgT,
> >
> > We would deeply appreciate it if you could kindly offer feedback on our rebuttal, as the rebuttal period will be closing in about one day. Your opinion would be invaluable to us, and we are sincerely grateful for your time and consideration.
> >
> > Thank you very much.
> >
> >
> > Best regards,
> >
> > The Authors of Paper 9690

---

### Official Review · Reviewer_M76W · 2024-11-03

**Soundness:** 2
**Presentation:** 2
**Contribution:** 2
**Rating:** 3
**Confidence:** 4

**Summary:**

This paper presented benchmark results on 6 existing public datasets and one newly collected industrial dataset with 12 recommendations models. There is an emphasis on multi-scenario recommendation, where different datasets are treated as multi-scenario by using some features for differentiating scenarios, such as user age, ad position, context, item category, platform, and news genres. Results are reported for how the number of “scenarios” affect model performance.

**Strengths:**

comprehensive experiments are done with a wide range of algorithms and datasets, including a newly collected industrial dataset, with code provided and clear instructions.

**Weaknesses:**

The contribution and novelty is limited since the authors merely present existing public benchmark datasets with some features for differentiating “scenarios”, whereas some, if not all, of these features could very well be just normal features, and no justification is provided on why it’s a reasonable choice to make them “scenario” features, and it seems the results are not benchmarked with treating the dataset as “single-scenario” as is, and treat the “scenario-feature” as normal feature. How the "scenario" features are handled are also not clearly described.

Many experiments are done, but limited insights are drawn from these experiments except some observations. Especially how the “scenarios” are modeled, and why certain models should be better than others.

The introduction of the industrial dataset seems new, but the description seems quite plain, and it’s not convincing why this newly collected dataset is a good dataset for benchmarking.

**Questions:**

Why does it make sense to artificially make certain features differentiate scenarios?

What is the “301” context feature?

What is “dense scenario” and “sparse scenario”?

**Details Of Ethics Concerns:**

this paper collected user data from an online advertisement platform. The paper claim to have annoymized user-identity the technical preprocessing of the data (e.g., hashing etc.), but probably need double check privacy concerns when it's published.

---

> ### Author Response · Authors · 2024-11-18
> **Feedback to Reviewer M76W - 1/2**
>
> Dear Reviewer M76W,
>
> We thank you for your thoughtful comments. We believe there might be some misunderstandings due to certain sections being overlooked. Below, we kindly provide clarifications.
>
> **W1: The contribution and novelty is limited since the authors merely present existing public benchmark datasets with some features for differentiating “scenarios”...**
>
> **A1**: We appreciate your comment and kindly ask you to revisit Appendix A.3.1, where we provide a detailed discussion of the scenario-splitting strategies. We would like to emphasize that we do not arbitrarily choose normal features for splitting. Instead, we select features based on well-recognized context features or those that have been previously used in prior works for scenario splitting. This approach ensures that our benchmark is grounded in established methodologies.
>
> - Ali-CCP and KuaiRand are designed for multi-scenario recommendation tasks, with context features like "301" in Ali-CCP representing the recommendation position, and "tab" in KuaiRand indicating the recommendation page or main page of the app.
> - Amazon and Douban datasets naturally involve interactions from different categories(platforms), making them ideal for MSR tasks. These datasets have been used extensively in prior research (e.g., [1, 2, 3]).
> - Recent works, such as [4], have demonstrated that scenario splits based on features beyond just context features can also enhance performance. Hence, our benchmark follows the previous works [5, 6], including MovieLens and Mind, which utilize user- and item-feature splitting (user feature "age" and item feature "genres").
>
> However, the question you mentioned is indeed an area of research that remains underexplored. Selecting the optimal feature for scenario splitting continues to be a challenging task in MSR, particularly in industrial applications that involve thousands of features. Works such as [4, 7], which investigate automatic feature selection for scenario splitting, are discussed.
>
> Regarding "Single-scenario" modeling, our benchmark is based on constructing "multi-scenarios" for a dataset to enable a fair comparison of model performance across different scenarios. This approach explores domain collaboration and specific processes, distinguishing it from comparisons in a "single-scenario" setting.
>
> For "scenario" feature handling, we clarify that firstly, scenarios are split based on the values of "scenario" features. Secondly, different methods process scenario features in varied ways—some, like EPNet, treat them in isolation, embed it isolatatedly, while others, such as HAMUR and SAR-Net, concatenate it with other features (e.g., user and item features) for unified embedding.
>
>
> [1]. Multi-domain Recommendation with Embedding Disentangling and Domain Alignment. CIKM'23.
>
> [2]. PLATE: A Prompt-Enhanced Paradigm for Multi-Scenario Recommendations. SIGIR'23.
>
> [3]. Diff-MSR: A Diffusion Model Enhanced Paradigm for Cold-Start Multi-Scenario Recommendation. WSDM'24.
>
> [4]. D3: A Methodological Exploration of Domain Division, Modeling, and Balance in Multi-Domain Recommendations. AAAI'24.
>
> [5]. HAMUR: Hyper Adapter for Multi-Domain Recommendation. CIKM'23.
>
> [6]. Multi-Domain Multi-Task Mixture-of Experts Recommendation Framework. SIGIR'24.
>
> [7]. DFFM: Domain Facilitated Feature Modeling for CTR Prediction. CIKM'23.
>
>
> **W2: Many experiments are done, but limited insights are drawn from these experiments except some observations. Especially how the “scenarios” are modeled, and why certain models should be better than others.**
>
>
> **A2**: We sincerely thank you for your insightful comments. However, we would like to clarify that our work goes beyond merely providing observations. We also offer a detailed analysis of leading models and provide the rationale behind each dataset in Appendix C. Additionally, we would like to emphasize that, unlike previous MSR research, we introduce Section 5.1.3-Scenario Number Analysis, where we not only present key observations but also explain the underlying reasons, examining how changes in scenarios affect performance, a perspective that, to the best of our knowledge, has not been explored in previous work.
>
> We believe that our contributions provide significant insights for both MSR scholars and industrial researchers. Based on our implementation and extensive results—including performance metrics, efficiency evaluations, scenario-specific analyses, and implementation details, readers will gain valuable understanding and actionable takeaways from our benchmark.

---

> ### Author Response · Authors · 2024-11-18
> **Feedback to Reviewer M76W - 2/2**
>
> **W3: The introduction of the industrial dataset seems new, but the description seems quite plain, and it’s not convincing why this newly collected dataset is a good dataset for benchmarking.**
>
> **A3**: We would like to thank you for recognizing our contribution of industrial dataset. and we will explain why we chose to use an industrial dataset for further validation.
>
> Firstly, there is a significant gap between real-world recommendation systems and publicly available datasets in MSR. Real-world recommendation systems often involve tens, or even hundreds, of features that are not present in public datasets. Our industrial dataset, with 108 features across 10 different scenarios, offers a more realistic and comprehensive context for evaluation, which is unmatched by other public datasets.
>
> Secondly, our benchmark is designed not only for research purposes but also as a practical tool for industrial companies to conduct real-world MSR testing. Since MSR is directly related to click rates, which in turn are closely linked to revenue, reliable, real-world testing is crucial to ensure the relevance and applicability of our findings.
>
> Lastly, we would like to highlight that our industrial dataset is derived from real-world user logs, making it particularly well-suited for multi-scenario research and exploration, especially when compared to other MSR datasets.
>
> **Q1: Why does it make sense to artificially make certain features differentiate scenarios?**
>
> **A4**: Kindly refer to A1. We have split the scenarios based on the specific meanings of these features or by following approaches from previous works, rather than subjectively determining the feature segmentation for each scenario.
>
> **Q2: What is the “301” context feature?**
>
> **A5**: We kindly invite you to refer to [Aliccp](https://tianchi.aliyun.com/dataset/408). The "301" feature refers to the "position" where the interaction occurred, with three distinct values corresponding to three different scenarios.
>
> **Q3: What is “dense scenario” and “sparse scenario”?**
>
> **A6**: We apologize for the confusion caused. We would like to provide a clearer explanation of these technical terms in MSR. "Dense scenarios" generally refer to interactions that occur more frequently, which we term as "dense." Conversely, "sparse scenarios" refer to scenarios where interactions are relatively rare, which we call "sparse." For example, in the case of Aliccp, S-1 is a "sparse scenario", while S-0 and S-2 are considered "dense scenarios".
>
> | ALI-CCP|S-0|S-1|S-2|
> |-|-|-|-|
> |# Interaction|32,236,951|639,897|52,439,671|
>
>
> We hope this explanation clarifies our approach, and these explanations could help you understanding the contribution of our benchmark.

---

> > ### Author Response · Authors · 2024-11-24
> > **Look forward to your feedback**
> >
> > Dear Reviewer M76W,
> >
> > We sincerely thank you for your insightful feedback and questions. During the discussion period, we provided clarifications and made modifications to address your concerns, summarized as follows:
> >
> > * Clarified the rationale behind the selection of the "scenario feature" in dataset processing.
> > * Provide feedback regarding insufficient analysis of claims.
> > * Emphasized the necessity of validation using industrial datasets.
> > * Provided explanations regarding artificial splitting scenarios and specific technical terms.
> >
> > We would greatly appreciate it if you could let us know whether our responses have addressed your concerns and consider reevaluating our paper. We are keen to hear any further feedback you might have. We are looking forward to your insights!
> >
> > Best Regards,
> >
> > The Authors of Paper 9690

---

> > ### Author Response · Authors · 2024-12-01
> > **Look forward to your feedback**
> >
> > Dear revieres M76W,
> >
> > We would be truly grateful if you could kindly provide at least one round of feedback on our rebuttal, as the rebuttal period is closing in nearly one day. We greatly appreciate your time and consideration. Thank you.
> >
> > Best Regards,
> >
> > The Authors of Paper 9690

---

### Official Review · Reviewer_dKPL · 2024-11-04

**Soundness:** 3
**Presentation:** 3
**Contribution:** 3
**Rating:** 6
**Confidence:** 4

**Summary:**

The paper provides a much-needed benchmark for the emerging multi-scenario recommender systems (MSR) field. This benchmark contains a comprehensive framework (including training and evaluation pipelines) and inspects multiple public and industrial data sets. The detailed comparison of various MSR models provides valuable insights into their advantages and disadvantages. In addition, the source code is available to the public.

**Strengths:**

S1: The paper presents the first dedicated benchmark for multi-scenario recommendation tasks, which may become a valuable resource in the field. It offers a comprehensive pipeline that includes data processing, model training, and evaluation, setting a new standard for transparency and reproducibility in MSR research.

S2: Including public and industrial datasets strengthens the benchmark's reliability and applicability, covering many real-world scenarios. The publicly available source code and detailed tutorials will greatly facilitate researchers in conducting experiments and building upon the benchmark.

S3: The paper provides a fair and detailed comparison of twelve state-of-the-art MSR models, which is valuable for researchers looking to understand the current landscape of MSR.

**Weaknesses:**

W1: Although the advantages and disadvantages of different baselines are provided, it can further enhance insights by solving the theoretical basis of the model and providing mitigation strategies for the "seesaw effect."

W2: The focus of standard MSR is understandable, but it does not explore the benchmark testing of more scenario-related topics (e.g., multi-scenario multi-task) and additional information (e.g., user's interactive history sequence), limiting the scope of its practical procedures.

W3: The paper could benefit from a more detailed analysis of the models' computational efficiency, particularly how they scale with dataset size and complexity, which is crucial for practical deployment. In addition, moral considerations in the MSR system are also beneficial, especially data privacy and potential prejudice.

**Questions:**

Q1: Can the cause of different baselines on different data sets provide some additional insights? Are there any strategies that can be summarized for the seesaw effects?

Q2: Can the authors provide more detailed analyses or benchmarks on the computational efficiency and scalability of the MSR models, especially in resource-constrained environments?

Q3: Can the moral considerations in the MSR system provide some corresponding insights?

---

> ### Author Response · Authors · 2024-11-19
> **Feedback to Reviewer dKPL - 1/3**
>
> Dear Reviewer dKPL,
>
> We appreciate your acknowledgment of our contributions. In response to the weaknesses and questions you have mentioned, we would like to provide the following feedback.
>
> **W1: Although the advantages and disadvantages of different baselines are provided, it can further enhance insights by solving the theoretical basis of the model and providing mitigation strategies for the "seesaw effect".**
>
> **A1**: We sincerely appreciate your insightful comments regarding theoretical analysis. Indeed, theoretical analysis remains a significant challenge in MSR research and has yet to produce widely recognized results that have passed peer review and gained community consensus. However, there emerge some recent reserch exploring theoretical analysis within MSR. For instance, studies such as CausalInt [1] and M-scan [2] employ causal analysis to investigate causal relationships across scenarios. More recently, papers like DSFA [3] aim to systematically explain phenomena such as the "seesaw effect" from a distributional perspective across different scenarios. We will continue to monitor and review the latest research. Should any peer-reviewed and widely accepted theoretical analysis become available, we will promptly incorporate it into the benchmark.
>
> [1]. CausalInt: Causal Inspired Intervention for Multi-Scenario Recommendation. KDD'22.
>
> [2]. M-scan: A Multi-Scenario Causal-driven Adaptive Network for Recommendation. WWW'24.
>
> [3]. Retrievable Domain-Sensitive Feature Memory for Multi-Domain Recommendation. Arxiv'24.
>
>
> **W2: The focus of standard MSR is understandable, but it does not explore the benchmark testing of more scenario-related topics (e.g., multi-scenario multi-task) and additional information (e.g., user's interactive history sequence), limiting the scope of its practical procedures.**
>
> **A2**: We deeply appreciate your valuable suggestions regarding the extended content of benchmarks. We are actively monitoring advancements in multi-scenario, multi-task recommendation research. Notably, models currently included in our benchmark, such as M2M, M3oE, and PEPNet, are grounded in multi-scenario, multi-task paradigms. As part of our near-future roadmap, we plan to further update the benchmark to accommodate multi-scenario, multi-task settings.
>
> Regarding your mention of sequential tasks, we acknowledge that this is an area with relatively limited focus in current research. We will closely monitor advancements in this subfield and, if relevant studies emerge, we will expand our benchmark to incorporate sequential recommendation tasks.

---

> ### Author Response · Authors · 2024-11-19
> **Feedback to Reviewer dKPL - 2/3**
>
> **W3: The paper could benefit from a more detailed analysis of the models' computational efficiency, particularly how they scale with dataset size and complexity, which is crucial for practical deployment. In addition, moral considerations in the MSR system are also beneficial, especially data privacy and potential prejudice.**
>
> **A3**: Thank you for highlighting your concerns regarding computational efficiency and ethical considerations.
>
> As for computational efficiency, we would like to provide more analysis of computational efficiency:
> - Our dataset is designed to accommodate a wide range of research needs, spanning from small-scale to large-scale validation. For studies focused on small-scale validation or resource-constrained multi-scenario recommendation tasks, we recommend exploring datasets such as MovieLens, Amazon, and Douban. These datasets enable efficient experimentation, with model training typically completing within minutes and inference times per batch averaging under 5ms. For larger-scale scenarios, datasets such as Ali-CCP, Kuairand, and Mind may be more appropriate. In these cases, model training generally requires several hours, and inference times increase correspondingly, often exceeding 5ms per batch. It is important to note that the computational efficiency of models varies significantly as dataset size increases.
> - Regarding the changes observed in specific models, we would like to point out that most models exhibit a corresponding growth in parameter size, such as training and inference efficiency, as dataset size scales up. This trend is clearly illustrated in Table 4. Among these models, we would particularly highlight M2M and Hamur for their "dynamic parameter mechanisms", which making them to adapt effectively to varying scenarios. M2M, in particular, demonstrates exceptional performance, as shown in Table 3. However, it is worth noting that as datasets scale up, these models also experience the largest increase in parameter size and inference time, aligning with the "no free lunch" principle. Specifically, M2M has a longer inference time due to its larger parameter size and the meta-unit mechanism it employs.
>
> With respect to ethical considerations, our benchmark focuses primarily on addressing concerns related to industrial datasets, as detailed in Section 6.2 of our work. Multi-scenario settings inherently involve user behavior across diverse platforms and contexts,presenting a higher risk related to privacy leakage or biases. To mitigate these risks, we strictly adhere to user consent protocols during data collection and implement stringent measures to ensure trustworthiness. Specifically, all sensitive user features are either removed or irreversibly encoded to guarantee anonymity, maintaining trustworthiness as our top priority.

---

> ### Author Response · Authors · 2024-11-19
> **Feedback to Reviewer dKPL - 3/3**
>
> **Q1: Can the cause of different baselines on different data sets provide some additional insights? Are there any strategies that can be summarized for the seesaw effects?**
>
> **A4**: We thank you for your valuable suggestions. Here, we would like to provide additional insights into the reasons behind the varying performance of different baseline models across diverse datasets and analysis of migrating seesaw effects.
>
> - First, please revisit Table 7 in the appendix. Different datasets adopt distinct scenario partitioning strategies, which lead to significant differences in information sharing (e.g., users, items) paradigm. We believe this variation has a notable impact on the performance of multi-scenario baseline models across different datasets, as the strategies for multi-scenario information extraction and scenario collaboration can vary substantially.
> - Specifically, as shown in Table 3, under datasets with the same partitioning strategy, such as KuaiRand and Aliccp, M2M and SAR-Net consistently achieve better results, while ADL and SharedBottom consistently underperform slightly. In contrast, on datasets like Amazon, Douban, and Mind, which share user information, SAR-Net also demonstrates strong and consistent performance, but M2M does not consistently maintain its superior results. This observation aligns with our earlier point that scenario-level information sharing significantly influences the outcomes.
> - Regarding the mitigation of the "seesaw effect", based on our experimental results, particularly from the SCENARIO NUMBER EXPERIMENT in Section 5.1.3, we believe that the key strategy lies in the design of the model architecture, especially the method of parameter sharing. For models like SharedBottom and STAR, they utilize a direct parameter-sharing structure, which leads to limited flexibility and poor generalization when extending to new domains, resulting in suboptimal performance. In contrast, models such as SAR-Net and HAMUR adopt an implicit parameter-sharing architecture in their domain-sharing design. By employing mechanisms like Debias-net or the Behavior Extract Layer, they implicitly share information across domains, demonstrating superior performance in domain extension scenarios. Therefore, to effectively address the "seesaw effect", adopting an implicit sharing approach rather than direct sharing for handling cross-scenario information is a more promising strategy.
>
>
>
> **Q2: Can the authors provide more detailed analyses or benchmarks on the computational efficiency and scalability of the MSR models, especially in resource-constrained environments?**
>
> **A5**: We kindly ask you to refer to our response in A3, where we provided more detailed analyses and benchmarks on the computational efficiency and scalability of the MSR models. Additionally, under resource-constrained scenarios, we recommend starting with smaller datasets (Amazon, Douban, MovieLens) and their corresponding models for initial validation. This approach ensures minimal resource consumption and allows for faster iterative verification. When scalability validation is required, we suggest extending the experiments to larger datasets (KuaiRand, Aliccp, Mind) to assess the models' performance at scale.
>
>
> **Q3: Can the moral considerations in the MSR system provide some corresponding insights?**
>
> **A6**: We encourage you to refer to our response in A3, where we highlighted the moral considerations in the benchmark. Addressing your concerns, we believe that MSR tasks require stricter user privacy protections compared to other recommendation tasks. This is because MSR systems often involve cross-platform user behavior data, increasing the risk of privacy leakage. Therefore, implementing robust measures such as encrypting privacy-sensitive features, protecting sensitive information, and adopting other precautions is essential to mitigate these risks. These steps are crucial for building more trustworthy systems and ensuring safer applications.

---

> > ### Author Response · Authors · 2024-11-24
> > **Look forward to your further feedback**
> >
> > Dear Reviewer dKPL,
> >
> > We sincerely appreciate the time and effort you have dedicated to reviewing our work, as well as your recognition of its contributions, which is truly valuable to us. As the discussion period is coming to an end, we kindly ask if you have any additional comments or thoughts on whether we have adequately addressed your concerns.
> >
> > Thank you once again for your constructive feedback, time, and patience throughout this process.
> >
> > Best regards,
> >
> > The Authors of Paper 9690

---

> > > ### Comment · Reviewer_dKPL · 2024-12-02
> > >
> > > I thank the authors for their detailed response, which addressed most of my concerns.
> > >
> > > I hope the authors can incorporate all reviewers' comments to improve this promising paper.

---

> > > > ### Author Response · Authors · 2024-12-02
> > > >
> > > > Dear Reviewer dKPL,
> > > >
> > > > We sincerely appreciate your recognition of our paper and are pleased to have addressed most of your concerns in our rebuttal. We promise to incorporate all the comments from all reviewers to further enhance the quality of our paper.
> > > >
> > > > Thank you once again for your time and patience.
> > > >
> > > > Best regards,
> > > >
> > > > The Authors of Paper 9690

---

### Note · Authors · 2024-12-04

**Comment:**

We thank the reviewers for their efforts in reviewing our paper. After careful consideration, we have decided to withdraw the paper for further refinement.

**Withdrawal Confirmation:**

I have read and agree with the venue's withdrawal policy on behalf of myself and my co-authors.